# Vertical structure of the lower-stratospheric moist bias in the ERA5 reanalysis and its connection to mixing processes

Konstantin Krüger[1], Andreas Schäfler[1], Martin Wirth[1], Martin Weissmann[3], and George C. Craig[2]

[1]Deutsches Zentrum für Luft- und Raumfahrt (DLR), Institut für Physik der Atmosphäre, Oberpfaffenhofen, Germany
[2]Meteorologisches Institut München, Ludwig-Maximilians-Universität, Munich, Germany
[3]Institut für Meteorologie und Geophysik, Universität Wien, Vienna, Austria

*Correspondence to*: Konstantin Krueger (konstantin.krueger@dlr.de)

**Abstract**

Numerical weather prediction (NWP) models are known to possess a distinct moist bias in the midlatitude lower stratosphere which is expected to affect the ability to accurately predict weather and climate. This paper investigates the vertical structure of the moist bias in the European Centre for Medium-Range Weather Forecasts (ECMWF) latest global reanalysis ERA5 using a unique multi-campaign data set of highly-resolved water vapour profiles observed with a differential absorption lidar (DIAL) onboard the High Altitude and LOng range research aircraft (HALO). In total, 41 flights in the midlatitudes from six field campaigns provide roughly 33000 profiles with humidity varying by four orders of magnitude. The observations cover different synoptic situations and seasons and thus are suitable to characterize the strong vertical gradients of moisture in the upper troposphere and lower stratosphere (UTLS). The comparison to ERA5 indicates high positive and negative deviations in the UT, which on average lead to a slightly positive bias (15–20 %). In the LS, the moist bias rapidly increases up to a maximum of 55 % at 1.3 km altitude above the thermal tropopause (tTP) and decreases again to 15–20 % at 4 km altitude. Such a vertical structure is frequently observed, although the magnitude varies from flight to flight. The layer depth of increased moist bias is smaller at high tropopause altitudes and larger when the tropopause is low. Our results also suggest a seasonality of the moist bias, with the maximum in summer exceeding fall by up to a factor of 3. During one field campaign, collocated ozone and water vapour profile observations enable a classification of tropospheric, stratospheric and mixed air using water vapour–ozone correlations. It is revealed that the moist bias is high in the mixed air while being small in tropospheric and stratospheric air, which highlights that excessive transport of moisture into the LS plays a decisive role for the formation of the moist bias. Our results suggest that a better representation of mixing processes in NWP models could lead to a reduced LS moist bias that, in turn, may lead to more accurate weather and climate forecasts. The lower-stratospheric moist bias should be borne in mind for climatological studies using reanalysis data.

# 1 Introduction

Water vapour is one of the most important greenhouse gases in the atmosphere and plays a key role for accurately predicting
the Earth's weather (Gray et al., 2014; Shepherd et al., 2018) and climate (Forster and Shine, 2002; Riese et al., 2012). In the
upper troposphere and lower stratosphere (UTLS), defined as a layer located ± 5 km around the thermal tropopause (tTP)
(Gettelman et al., 2011), rapidly decreasing water vapour concentrations in the vertical (e.g., Kiemle et al., 2012; Kaufmann
et al., 2018) are of key relevance to a net cooling near and above the tropopause (Randel et al., 2007). The radiative modulation
of the vertical temperature gradients may influence the near tropopause potential vorticity (PV) gradient (Chagnon et al., 2013)
that acts as a waveguide for Rossby waves (Martius et al., 2010) and thus may affect downstream weather development in the
midlatitudes. Hence, an accurate representation of UTLS water vapour in numerical weather prediction (NWP) and climate
models is essential.

In the extratropical UTLS, the distribution of water vapour is driven by transport and mixing processes related to baroclinic
waves and associated synoptic and meso-scale weather systems, which are interacting with chemical processes (e.g., Gettelman
et al., 2011; Schäfler et al., 2022). The increased static stability above the tropopause (Birner et al., 2002) impedes water
vapour from being vertically transported. Correspondingly, the sharpest decline of water vapour is found just above the
tropopause. Exchange processes affect the water concentration around the tropopause (Holton et al., 1995; Stohl et al., 2003)
and create the extratropical transition layer (ExTL; Pan et al., 2004; Hoor et al., 2010) with influences of the troposphere and
the stratosphere. In particular quasi-isentropic exchange near the polar and subtropical jet streams (Haynes and Shuckburgh,
2000) and cross-isentropic mixing, for instance through overshooting convection (e.g., Dessler and Sherwood, 2004; Homeyer
et al., 2014), are major contributors to increased humidity above the tropopause. Furthermore, tropopause folds are related to
mass exchange between the UT and the LS (Shapiro et al., 1980). Above the ExTL, the concentration of water vapour
approaches a low and vertically constant background value (e.g., Hintsa et al., 1994), which is determined by the stratospheric
transport from tropics (Fueglistaler et al., 2009) within the Brewer-Dobson circulation (e.g., Dobson et al., 1946; Brewer,
1949) on time scales from months to years (Birner and Bönisch, 2011). The complexity of transport and mixing processes is
mirrored in the high water vapour variability in the extratropical UTLS on synoptic and seasonal time scales (e.g., Pan et al.,
2000; Randel and Wu, 2010; Zahn et al., 2014; Dyroff et al., 2015; Bland et al., 2021; Schäfler et al., 2022).

The sharp vertical gradients of trace species, PV, wind and temperature at the extratropical tropopause are challenging to
resolve for state-of-the-art NWP models (e.g., Stenke et al., 2008; Schäfler et al., 2020). Current NWP analyses and forecasts
are known to possess a distinct moist bias in the extratropical LS (e.g., Kaufmann et al., 2018) which is causing a collocated
cold bias at the same altitudes (Stenke et al., 2008; Diamantakis and Flemming, 2014; Shepherd et al., 2018). Recently, Bland
et al. (2021) used radiosonde observations of a two-month period in fall and confirmed the earlier documented moist bias
(about 70 % in the LS) in current operational analyses of the European Centre for Medium-Range Weather Forecasts
(ECMWF) Integrated Forecast System (IFS) and the Met Office's Unified Model (METUM). They also showed that radiative
effects related to the moist bias cause a collocated cold bias in the LS that is growing with forecast lead time. For a

comprehensive overview of the studies that quantified the LS moist bias in different NWP systems, the interested reader is referred to Table 1 in Bland et al. (2021). The vertical structure of the moist bias is characterized by a small positive bias below the thermal tropopause followed by a vertical increase in the LS to a maximum at 1–2 km above the tropopause (e.g., Dyroff et al., 2015; Bland et al., 2021). However, different shapes of the LS moist bias above its maximum have been reported. Bland et al. (2021) show an opposing vertical structure of the moist bias beyond 2 km above the tTP using two different radiosonde types. Woiwode et al. (2020) compare humidity cross sections of an airborne passive infrared imager and present a case with vertically increasing, one with constant and two cases with vertically decreasing moist bias in IFS analysis and forecast data. The origin of the wet model bias is still under debate: One hypothesis is, that it is caused by misrepresented dynamical transport and mixing processes (Kunz et al., 2014; Shepherd et al., 2018), e.g., overshooting convection leading to excessive water vapour injection into the LS. Another potential source of overestimated transport of moisture into the LS is numerical diffusion and insufficient model resolution in the semi-Lagrangian advection scheme used in the ECMWF model leading to an excessive diffusive transport of moisture across strong gradients from high to low mixing ratios (Stenke et al., 2008; Kunz et al., 2014; Dyroff et al., 2015; Shepherd et al., 2018). However, a LS moist bias of similar order is also found for "Eulerian"-formulated models (Jiang et al., 2015; Davis et al., 2017). Moreover, Woiwode et al. (2020) confirm that the bias is already present in the initial conditions and demonstrate a low response of the moist bias to variable vertical or temporal resolutions.

The above-mentioned studies used a variety of observation techniques to quantify the moist bias. Radiosonde or dropsonde humidity observations provide temporally continuous series of profiles at the same location, but their reliability is limited > 2 km above the tTP (e.g., Bland et al., 2021). In situ aircraft observations, even though very accurate and highly resolved, provide profile information only during start and landing and on flight routes of commercial or research aircraft (Zahn et al., 2014; Kunz et al., 2014; Dyroff et al., 2015; Kaufmann et al., 2018). On the contrary, spaceborne microwave sounder provide humidity information across the entire globe but are limited in vertical resolution (e.g., Hegglin et al., 2013; Jiang et al., 2015; Khosrawi et al., 2018). In between the in situ and satellite observations, profile data from active and passive remote sensing instruments onboard research aircraft demonstrated the potential to characterize humidity across the tropopause (Ehret et al., 1999; Flentje et al., 2007; Woiwode et al., 2020; Schäfler et al., 2021), combining high spatial coverage, high accuracy and high vertical resolution (Bhawar et al., 2011). Since 2013, the active Differential Absorption Lidar (DIAL) WAter vapour and Lidar Experiment in Space (WALES; Wirth et al., 2009) has been deployed in several research campaigns onboard the HighAltitude and LOng range research aircraft (HALO; Krautstrunk and Giez, 2012) for water vapour profile measurements. The goal of this paper is to evaluate the LS moist bias in the ECMWF's most recent global reanalysis ERA5. The model analyses are compared against a comprehensive data set of water vapour profiles observed by the airborne DIAL WALES in the midlatitude UTLS. Collocated water vapour and ozone profiles are used to identify tropospheric, stratospheric and mixed air and to individually assess the moist bias as we suspect that mixing processes affect the vertical structure of the moist bias. The following three specific questions are addressed:

1. Can the multi-campaign DIAL data set robustly quantify the LS moisture bias in ERA5?

2. What is the vertical structure of the LS moist bias in ERA5, particularly at high altitudes?

3. Is the moist bias correlated to the distribution of mixed air masses in the UTLS?

This paper is outlined as follows: Section 2 provides an overview of the water vapour DIAL observations (Sect. 2.1), the ERA5 reanalysis (Sect. 2.2) and the methods utilized to compare the observational and model data (Sect. 2.3). In Sect. 3.1 an example cross section of specific humidity and the bias are illustrated for a midlatitude jet stream crossing which is followed by a statistical tropopause-relative evaluation of the vertical structure of the bias and its variability in Sect. 3.2. The relationship between the vertical structure of the moist bias and the distribution tropospheric, stratospheric and mixed air is presented in Sect. 3. Thereafter, Sect. 4 provides a discussion of the results. The key conclusions are summarized in Sect. 5.

## 2 Data and Methods

### 2.1 The WALES data set

The DIAL WALES (Wirth et al., 2009) was developed at the German Aerospace Center (DLR) and has been operated onboard the German research aircraft HALO since 2010. The instrument design is based on two identical laser systems that generate four wavelengths in the near-infrared (NIR) absorption band of water vapour between 935 and 936 nm allowing water vapour observations from the planetary boundary layer up to the stratosphere. WALES furthermore operates two polarization-sensitive channels at 1064 nm and at 532 nm. The latter channel comprises of a high spectral resolution lidar (HSRL; Esselborn et al., 2008) enabling extinction coefficient observations and thus aerosol characterization (Groß et al., 2013). WALES and its underlying DIAL technique is briefly introduced in the following and a more detailed description can be found in Wirth et al. (2009).

The four NIR wavelengths are separated into three online channels (strongly absorbed by water vapour) and one offline channel (weakly absorbed). The number concentration of water vapour in the probed volume is derived from the ratio of the backscattered light of the on- and offline wavelengths and then converted to specific humidity. The online channels are sensitive to different trace gas concentrations and in turn to different altitude levels. The exact wavelengths are selected such that they are optimally aligned to the moist boundary layer, the UT and the dry LS. Note, that the WALES measure humidity profiles are only available in cloud-free regions or regions with optically thin clouds. In optically thick clouds the extinction by cloud particles is so strong that no water vapour information can be retrieved within or below the cloud.

Due to the photon statistics of the backscattered light as well as detector and background light noise, the retrieved water vapour profiles undergo statistical variations which are effectively reduced by temporal (i.e., horizontal) and vertical averaging. Thus, the retrieved DIAL water vapour profiles are averaged over 12 s or approximately 3 km in the horizontal. In the vertical, data is available every 15 m, although the effective vertical resolution is 300 m according to the full width of half maximum of the averaging kernel. It should be stressed, that the averaging kernel of the WALES DIAL is exactly zero outside of about $\sqrt{2}$

125 times the effective resolution. This is in sharp contrast to most passive remote sensing techniques where the side modes of the kernels can lead to erroneous dry or wet layers in the retrieved humidity profile. In the DIAL data retrieval, the statistical error of the observed volume is different for each flight and depends on the water vapour distribution and the background light. To remove high noise, typically occurring in dry air lying underneath moist air, e.g., in the vicinity of stratospheric intrusions (Trickl et al., 2016), we filtered 5 % of the noisiest data for each individual flight. This threshold turned out to be useful,

however, reduced the data availability in the lower-to-mid troposphere. Furthermore, Rayleigh-Doppler beam broadening, laser spectral impurity and uncertainties in spectral databases are sources for systematic errors, which are compensated for in the retrieval algorithm. The total systematic error was found to be in the order of 5 % (Kiemle et al., 2008). The high reliability of WALES was demonstrated in various intercomparisons, e.g., with Lyman-alpha in situ hygrometers (Kiemle et al., 2008), comparable airborne and ground-based DIAL instruments (Bhawar et al., 2011) and radiosondes with a frost point hygrometer

(Trickl et al., 2016).

In this study, we use DIAL observations from six campaigns from 2013–2021 that provide almost 33000 water vapour profiles obtained during 41 research flights. The profiles were sampled along the flight track and extend from the surface up to about 14 km altitude corresponding to the maximum flight level of the HALO aircraft (Krautstrunk and Giez, 2012). As the focus of this study is the midlatitude UTLS, we only consider flights that provide a significant amount of data across the tropopause.

The majority (25) of these flights took place in the northern hemispheric fall season during the North Atlantic Waveguide Downstream impact EXperiment (NAWDEX; Schäfler et al., 2018) and the Wave-driven ISentropic Exchange campaign (WISE; Kunkel et al., 2019). As part of the campaigns ElUcidating the RolE of Cloud-Circulation Coupling in ClimAte (EUREC[4]A; Stevens et al., 2021), the Next-generation Aircraft Remote sensing for VALidation studies (NARVAL; Klepp et al., 2014) and NARVAL2 (Stevens et al., 2019) measurements were taken during eight flights in winter season. In addition,

the Cirrus in High-Latitudes (CIRRUS-HL) mission provides observations in summer. Figure 1 depicts the parts of HALO research flights where DIAL observations were obtained. Most flights were carried out over the North Atlantic between 48 °N and 66 °N, the North Sea and central to western Europe. Additionally, the subtropics (> 35 °N) and the Arctic were covered by individual flights as well.

During the WISE campaign, WALES was operated in a different setup to measure both water vapour and ozone, concurrently.

For this purpose, two of the 935 nm NIR water vapour channels were replaced by two ultraviolet (UV) channels covering the 300–305 nm ozone absorption line (Fix et al., 2019). The use of two instead of four channels per trace gas leads to a reduced vertical coverage which was optimized so that the selected NIR wavelengths cover the tropopause region. Increased statistical noise required averaging over a period of 24 s (~6 km horizontally) while the effective vertical resolution remains approximately 300 m (Fix et al., 2019).

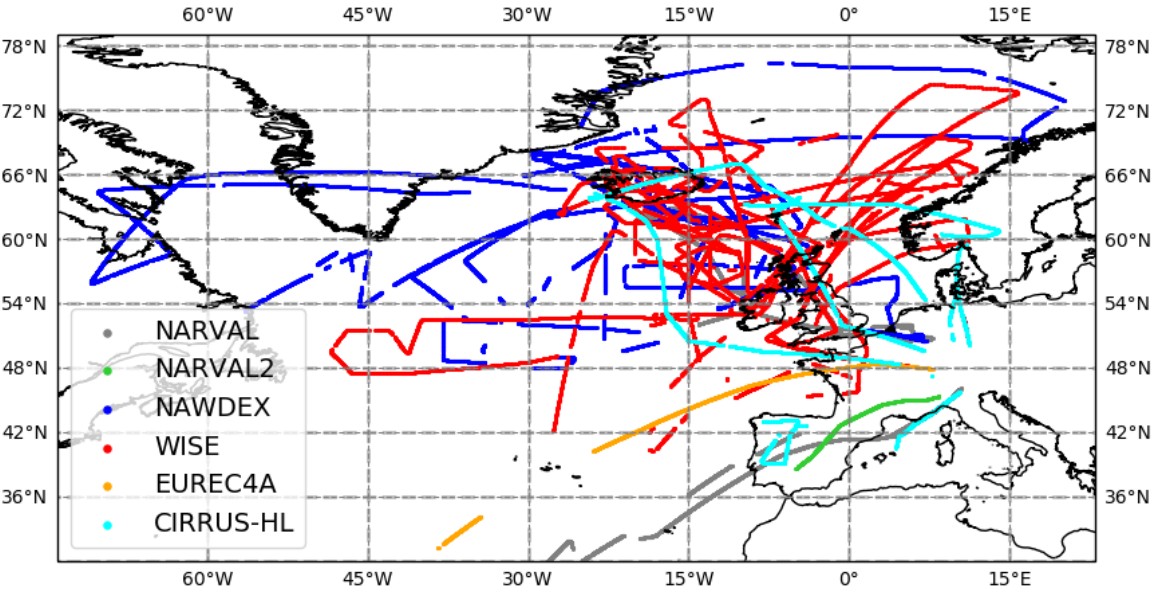

**Figure 1:** Map of HALO flight sections with WALES DIAL water vapour observations during the research campaigns NARVAL, NARVAL2, NAWDEX, WISE, EUREC⁴A and CIRRUS-HL (for detailed overview see Sect. 2.1).

The number of observations with respect to latitude (Fig. 2a) illustrates the high data availability in the midlatitudes, which is the region of interest in this study. This data set that covers humidity observations in a broad spectrum of synoptic situations is considered to be representative for midlatitude weather. Figure 2b shows the distribution of the water vapour observations covering four orders of magnitude ranging from $10^{-3}$ to $10^{1}$ g kg$^{-1}$. The bimodal shape of the histogram is composed of a broad moist part that can be assigned to the troposphere and a fraction of low humidity representing the dry conditions in the stratosphere. Each campaign exhibits an individual footprint of measured humidity, depending on the season, observation areas and the flight level selection. For instance, the histograms for NAWDEX and WISE are remarkably similar since both campaigns took place over the North Atlantic in fall. However, as only two NIR wavelengths were operated to measure water vapour during WISE, less measurements are available at high humidity levels. NARVAL shows a distinctive dry spectrum of measured humidity corresponding to the winter season and less data is available for the LS, resulting from frequent low flight altitudes. The CIRRUS-HL summer campaign stands out because a large proportion of high moisture values was observed. The NARVAL2 and the EUREC⁴A campaign provide UTLS measurements only for one flight and thus, compared to the other field campaigns, provide a small number of observations (see also Table 1).

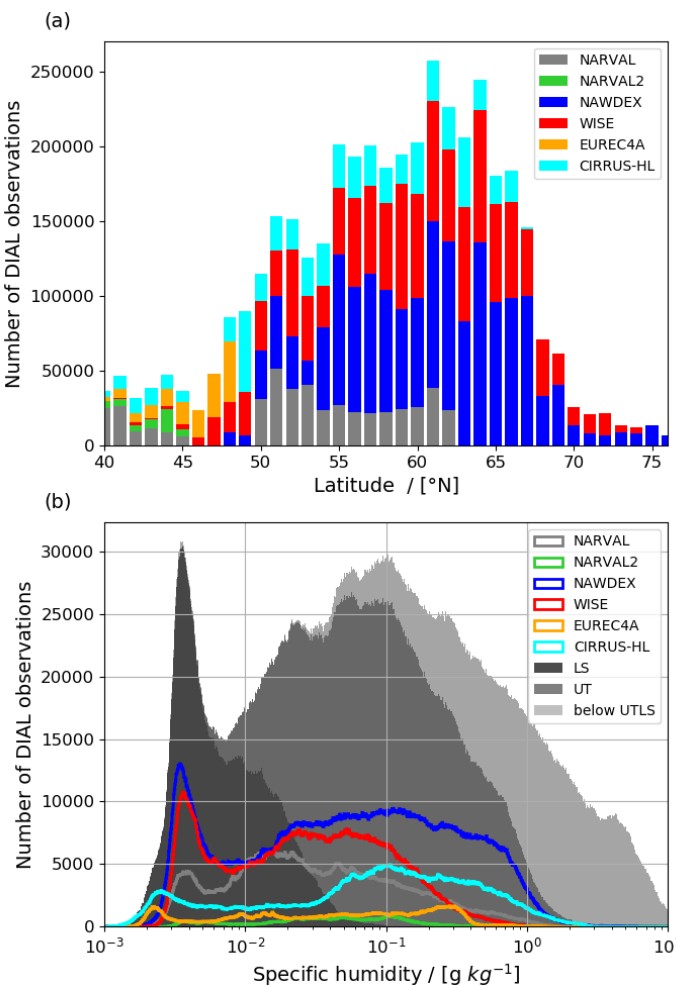

**Figure 2:** (a) Stacked distribution of the number of observations in 1° latitude bins per individual campaigns (coloured bars). (b) Histogram of observations per humidity bin with a size of 0.01 g kg$^{-1}$ of $\log_{10}$ ($q_{DIAL}$) for individual campaigns in the UTLS (coloured lines). Shading shows frequencies separated for the LS (all data above the thermal tropopause, dark grey shading), the UT (all data between the tTP and 5 km below, medium grey shading) and the remaining tropospheric data (light grey shading).

**Table 1**: Overview of all considered campaigns with DIAL observations. The number of DIAL profiles refers to all profiles that were sampled during 41 flights. The number of DIAL profiles in the LS corresponds to all profiles with measurements in the LS.

| Campaign | Year Month | Season | Number of Flights (hours) | Flight distance (km) | Number of DIAL profiles | Number of DIAL profiles in the LS |
|---|---|---|---|---|---|---|
| NARVAL | 2013/14 Dec–Jan | Winter | 7 (41) | 31157 | 10973 | 5288 |
| NARVAL2 | 2016 Aug | Summer | 1 (9) | 7729 | 2395 | 485 |
| NAWDEX | 2016 Sep–Oct | Autumn | 11 (75) | 55695 | 19139 | 12062 |
| WISE | 2017 Sep–Oct | Autumn | 14 (105) | 83041 | 13557 | 9493 |
| EUREC[4]A | 2020 Jan–Feb | Winter | 1 (8) | 7011 | 2307 | 1009 |
| CIRRUS-HL | 2021 Jun–Jul | Summer | 7 (30) | 23675 | 6777 | 4568 |
| Total | | | 41 (268) | 208308 | 55148 | 32905 |

## 2.2 ERA5 reanalysis data

ERA5 is the latest generation reanalysis of the ECMWF based on the IFS Cycle 41r2 that was used for operational weather prediction in 2016. Atmospheric quantities are provided on a global grid with a horizontal resolution (TL639) of about 31 km, and on 137 hybrid sigma-pressure model levels ranging from the surface up to 0.01 hPa (~80 km) in the vertical. The altitude range of the DIAL observations is covered by the lowermost 70 model levels. The vertical grid spacing of the model levels ranges from a few metres in the boundary layer to about 300 metres at the tropopause level (Schäfler et al., 2020). ERA5 reanalyses are available with a time resolution of one hour, which is an improvement compared to a six-hourly resolution of its predecessor ERA-Interim (Dee et al., 2011). Further details about ERA5 are documented in Hersbach et al. (2020). For this study, model level data is retrieved on a regular 0.36°x0.36° longitude/latitude grid. Pressure and altitude of each model level is derived following the IFS documentation (ECMWF, 2015). To be able to compare ERA5 and WALES data, the gridded model data is interpolated in space and time to the observation location. Our interpolation method uses a horizontally bi-linear interpolation, followed by a linear interpolation in the vertical. Finally, a linear interpolation in time of the hourly ERA5 profiles towards the observation time is carried out. This sequence of interpolation steps has been applied similarly in other studies (e.g., Schäfler et al., 2010).

**2.3 Data Processing**

**2.3.1 Thermal tropopause detection**

Due to the variable altitude of the tTP, the distribution of water vapour in the UTLS at individual altitudes is also highly variable. Hence, averaging of the humidity profiles in geometrical coordinates strongly blurs the vertical gradients across the tropopause. Therefore, bias statistics are often performed in tropopause-relative coordinates (e.g., Kunz et al., 2014; Bland et al., 2021). Different tropopause definitions have been established taking the thermal, dynamical and chemical properties of the
UTLS as a reference. By definition, the tTP marks the reversal of the vertical temperature gradient and thus the abrupt increase in static stability which is reflected in the sharp distribution of trace species across the tropopause (Gettelman et al., 2011). We use the tTP as it best reflects the strongest vertical gradients of water vapour (Birner et al., 2002; Pan et al., 2004). From each ERA5 temperature profile interpolated to the 15 m vertical grid of the lidar, we calculate the tTP altitude using the World Meteorological Organisation's (WMO) lapse rate-based definition (WMO, 1957). A tTP is detected as the lowest level at
which the vertical temperature gradient $\Gamma$ drops below 2 K km$^{-1}$ and is only defined if the average lapse rate between this and any other level within a 2 km deep layer remains equal or lower than 2 K km$^{-1}$. The vertical temperature gradient, i.e., the lapse rate, is computed as

$$\Gamma = \left(-\frac{dT}{dz}\right) [K \ km^{-1}]. \tag{1}$$

In our analyses, the tTP detection is started in upward direction from 5 km altitude in order to avoid misdetections of
tropopauses due to local fluctuations of temperature in the lower-to-mid troposphere. When a tTP is detected, the (thermal) tropopause-relative coordinates $z_{rel.tTP}$ are derived by simply subtracting the altitude of tTP ($z_{tTP}$) from the geometric height vector ($z_{geom}$):

$$z_{rel.tTP} = z_{geom} - z_{tTP}. \tag{2}$$

There are atmospheric conditions in which tropopause detection is ambiguous, especially in the vicinity of the jet streams and
associated tropopause folds where double tropopauses can occur (e.g., Shapiro, 1980; Gettelman et al., 2011). We found that in situations of weak vertical temperature gradients near the jet streams, the lapse rate threshold in the WMO definition may lead to vertical jumps of the tTP altitudes for adjacent profiles. These fluctuations result in a wrong vertical allocation of water vapour in tropopause-relative coordinates. A detailed discussion will follow in Sect. 3.1. To remove such profiles in the overall statistic, we apply a filtering method based on mean potential vorticity (MPV; Shapiro et al. 1998) which is the average PV
calculated for the 5 km layer above and below the thermal tropopause. MPV < 3.5 potential vorticity units (PVU; 1 PVU = 10$^{-6}$ K m² kg$^{-1}$ s$^{-1}$) above and MPV > 3.5 PVU below the tTP are found to be an efficient metric to filter profiles within an erroneously assigned tTP. Figure 3 shows the vertical distribution of tTP altitudes for the 32905 profiles which lies between 5.5 km and more than 15 km altitude, reflecting the broad spectrum of synoptic situations covered by the data set. The majority of all tTPs is found between 10 and 13 km which represents the typical location of the midlatitude tropopause with respect to
interannual or synoptic variations (e.g., Birner et al., 2002).

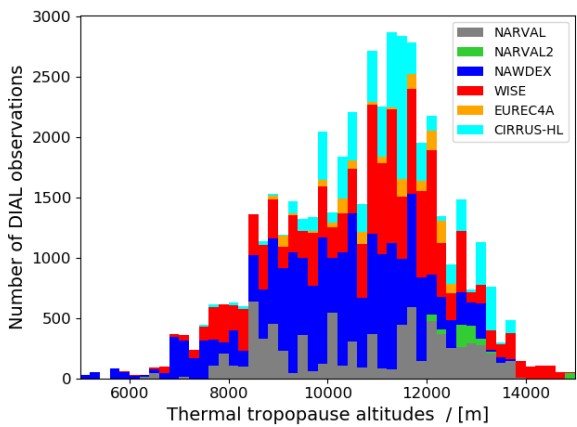

**Figure 3**: Histogram of the number of observations per thermal tropopause altitude bin (1000 m) and per campaign (coloured bars).

### 2.3.2 Statistical metric of the bias

The selection of a suitable difference metric is crucial for a robust quantification of model humidity errors and different
statistical approaches can be found in the literature (Kunz et al., 2014; Bland et al., 2021). As specific humidity rapidly decreases across the tropopause, absolute humidity differences are not appropriate and most studies rely on a relative formulation of the error. However, since the simple ratio of model and observation or the absolute bias divided by the observed value are statistical asymmetric quantities, we apply a logarithmic formulation with base 2 (see Eq. (3)) introduced by Kunz et al. (2014):

$$humidity\ bias\ = log_2\left(\frac{Q_{ERA5}}{Q_{DIAL}}\right), \tag{3}$$

with $Q_{DIAL}$ being the measured and $Q_{ERA5}$ being the ERA5 specific humidity. This unitless definition of the relative bias is symmetrically centred around zero and thus not distorted when averaged. A perfect agreement (humidity bias = 0) between the ERA5 and the DIAL specific humidity is reached if $Q_{ERA5} = Q_{DIAL}$. A positive humidity bias $\epsilon$ [0, ∞] indicates an overestimation of humidity by the model ($Q_{ERA5} > Q_{DIAL}$), whereas a negative humidity bias $\epsilon$ [-∞, 0] implies an
underestimation ($Q_{ERA5} < Q_{DIAL}$). Table 2 gives some example bias values for selected moisture observations.

**Table 2:** Some example values of specific humidity and the according computed humidity bias.

| $Q_{ERA5}$ | g kg⁻¹ | 0,50 | 0,75 | 1,00 | 1,25 | 1,50 | 1,75 | 2,00 | 2,25 | 2,50 | 3,00 |
|---|---|---|---|---|---|---|---|---|---|---|---|
| $Q_{DIAL}$ | g kg⁻¹ | 1,00 | 1,00 | 1,00 | 1,00 | 1,00 | 1,00 | 1,00 | 1,00 | 1,00 | 1,00 |
| **Humidity bias** | Unitless | -1,00 | -0.41 | 0 | 0,32 | 0,58 | 0,81 | 1,00 | 1,17 | 1,32 | 1,58 |
| **Percentage** | % | -50 | -25 | 0 | 25 | 50 | 75 | 100 | 125 | 150 | 200 |

**3 Results**

**3.1 Water vapour and bias distributions for a selected case**

First, an example cross section of water vapour measurements of the research flight on 1st October 2017 during the WISE campaign is presented in Fig. 4. The case is selected as it possesses a good data coverage across the UTLS and as it additionally provides ozone observations (see Sect. 3.3). HALO flew meridional transects over the North Atlantic (50 °N–60 °N) at 13 °W aiming to measure a zonal jet stream and its associated predicted strong trace gas gradients. The underlying synoptic situation and the corresponding mission objectives are provided in detail by Schäfler et al. (2021). The left part of Fig. 4a (up to a distance of roughly 800 km) illustrates the water vapour distribution north of the jet stream (see magenta isopleths) where the aircraft flew above the low-located tropopause within the LS. HALO then crossed the pronounced jet stream with wind velocities of more than 90 m s$^{-1}$. Near the jet core, the tTP altitude jumps from 6.5 to 14 km within a few kilometres flight distance. The dynamical tropopause (2 PVU contour line) also displays the ascent of the tropopause and a corresponding tropopause fold that extends along inclined isentropes into the mid-troposphere. In the right part of Fig. 4a, the air mass located to the south of the jet stream exhibits high tropopause altitudes exceeding the flight level by roughly 2 km, so that measurements are restricted to tropospheric air. Along the entire cross section, the highest specific humidity is observed at the lowest levels in the UT ranging from $10^{-2}$ g kg$^{-1}$ to occasionally more than 10 g kg$^{-1}$. The tropospheric air to the south of the jet stream has an increased humidity content compared to the air northward from the jet stream. In the LS, specific humidity values lower than $10^{-2}$ g kg$^{-1}$ are frequently observed. At a first glance, the specific humidity curtain of ERA5 (Fig. 4b) is very similar to the observations. However, the ERA5 humidity field appears to be smoother, particularly in the presence of strong horizontal water vapour gradients, for instance, near the jet stream and mesoscale filaments. Differences between observations and model, calculated by applying Eq. (3), are shown for the vertical section in Fig. 4c. Reddish regions indicate an overestimation of humidity by ERA5, while bluish areas represent an underestimation. High positive and negative values of the bias alternate below the tropopause. In the LS, a coherent region of positive values is detected between 1 and 3 km above the tTP indicating an overestimated humidity that extends over the entire part north of the jet. At the highest altitudes, beyond 3 km above the tropopause, the moist bias is smaller. In order to study the systematic nature of the diagnosed LS moist bias and its vertical structure, a statistic of all observations in tropopause-relative coordinates is performed.

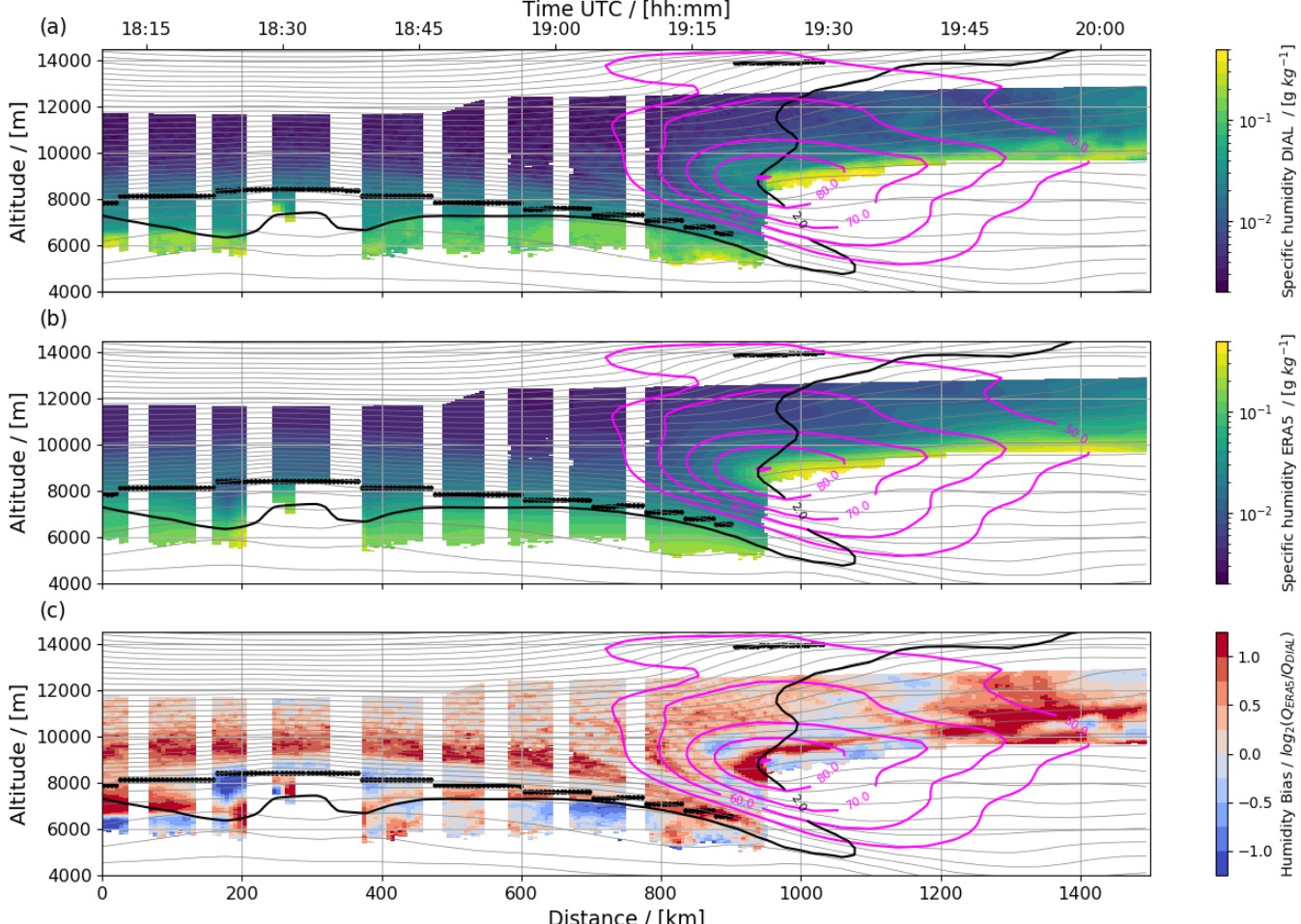

**Figure** 4: Vertical cross sections of (a) the DIAL specific humidity (colour shading, g kg$^{-1}$), (b) ERA5 specific humidity (colour shading, g kg$^{-1}$) as well as (c) the corresponding humidity bias (colour shading) on the 1$^{st}$ October 2017. (a) – (c) are superimposed by ERA5 fields of the potential temperature (grey contours, $\Delta\theta = 3K$) and the isopleths of the wind speed (magenta contours, in m s$^{-1}$), and the thermal (thick black dots) and the dynamical tropopause (2 PVU, black isoline).

## 3.2 Statistical analysis of the LS bias

### 3.2.1 Vertical structure

For all 32905 profiles from the 41 flights, the average profiles of specific humidity and the humidity bias are presented in Fig. 5. The moisture profiles of WALES and ERA5 show an exponential decline of specific humidity in the UT, ranging from about 5x10$^{-1}$ g kg$^{-1}$ at the lowest levels to approx. 3x10$^{-2}$ g kg$^{-1}$ at the tropopause. The strongest vertical gradient occurs in a layer of 0.5 to 1 km above the tropopause. Beyond, a less pronounced decline of water vapour extends until 4 km above the

tropopause followed by a vertical constant specific humidity of about 3.5x10$^{-3}$ g kg$^{-1}$. There is a high level of agreement

between the ERA5 and WALES specific humidity profiles, particularly in the UT, although ERA5 appears to be moister at all altitudes. For both data sets, the median and arithmetic mean profiles of specific humidity slightly vary from each other. The median line is slightly shifted towards drier humidity values, most pronounced in the UT. Figure 5a also demonstrates a high data availability throughout the entire UTLS. The number of observations is highest between -5 and 1 km around the tTP, with two local maxima at -2 km and roughly 1 km. Note that these two peaks in data availability are related to the typical flight altitudes, either above or below the main transatlantic air traffic routes (Schäfler et al., 2018) and the maximum data coverage close to the aircraft. Above the tTP, the number of observations continuously decreases and roughly halves per kilometre altitude. At 4 km above the tTP ~3000 observations are available.

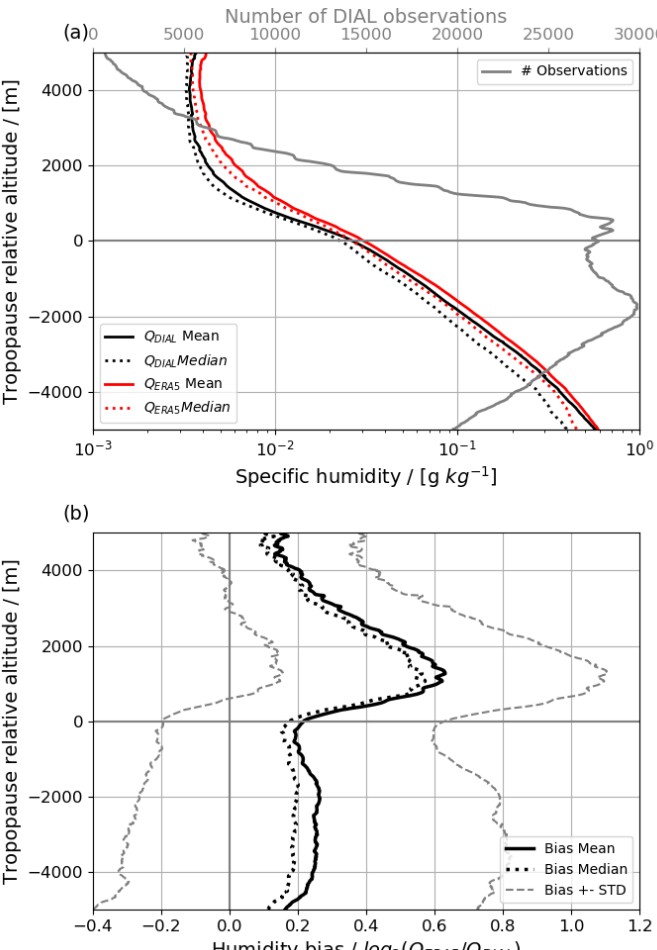

**Figure 5:** Tropopause-relative (a) vertical profiles of the DIAL (black lines) and the ERA5 (red lines) mean (solid) and median (dotted) specific humidity and the number of observations (grey). Note the log-scale notation of the x-axis. (b) Mean/median bias (solid/dotted lines) and standard deviation (grey dotted lines).

The higher moisture values in the ERA5 data become apparent in the vertical profile of the humidity bias (Fig. 5b) that is weakly positive (0.2; 15 %) in the UT and associated with a high standard deviation. This is a result of strong positive and negative bias values, as seen for example in the case study (Fig. 4c). The weakest bias of 0.2 (< 15 %) is reached at the tTP level. Above, the vertical moisture gradient is stronger in observations leading to a significant overestimation of humidity in the LS up to 4 km above tTP. The bias increases to a maximum of 0.63 (55 %) at 1.3 km altitude above the tTP. Beyond, the bias reduces by roughly 0.2 per 500 m up to 4 km altitude above the tTP, where it is approx. 0.2 (15 %). At the highest altitudes (> 4 km above the tTP) a weak and vertically nearly constant bias is observed. At the tropopause as well as above, the standard deviation is significantly reduced compared to the UT. Mean and median profiles of the humidity bias slightly differ, but these differences are very small compared to the magnitude of the bias. The maximum mean and median biases are 0.63 (55 %) and 0.58 (49 %), respectively.

To better illustrate the variability of the water vapour observations in the vertical, Fig. 6 shows the number of data and the mean bias in bins of tropopause-relative altitude and specific humidity. Figure 6a indicates a broader observed distribution of water vapour in the UT compared to the LS. A small number of unusually low humidity values ($< 10^{-2}$ g kg$^{-1}$) is detected below the tropopause (-4 to -1 km) and on the other hand some data show high specific humidity observations ($> 10^{-2}$ g kg$^{-1}$) are detected at approx. 1–3 km above the tTP. These observations are related to with incorrectly assigned tropopause altitudes that were not removed by the applied MPV filtering (see Sect. 2.3.1). However, these remaining outliers are tolerable as they have a negligible impact on the statistics. Throughout the UT, a weak positive bias is detected in bins of highest data availability. At the edges of the distribution, highest humidity values show a negative bias while the lowest humidity values stand out due to a positive bias (Fig. 6b). We found that this is related to a narrower distribution of ERA5 humidity compared to the observations (not shown). The low number of observations at the edges should be noted here. In the LS the positive bias is higher and most pronounced up to 3.5 km above the tropopause and at very low specific humidity values. The positive bias reduces towards highest altitudes (> 3 km above the tropopause) of the LS, although the reduced data coverage has to be kept in mind (Fig. 5a).

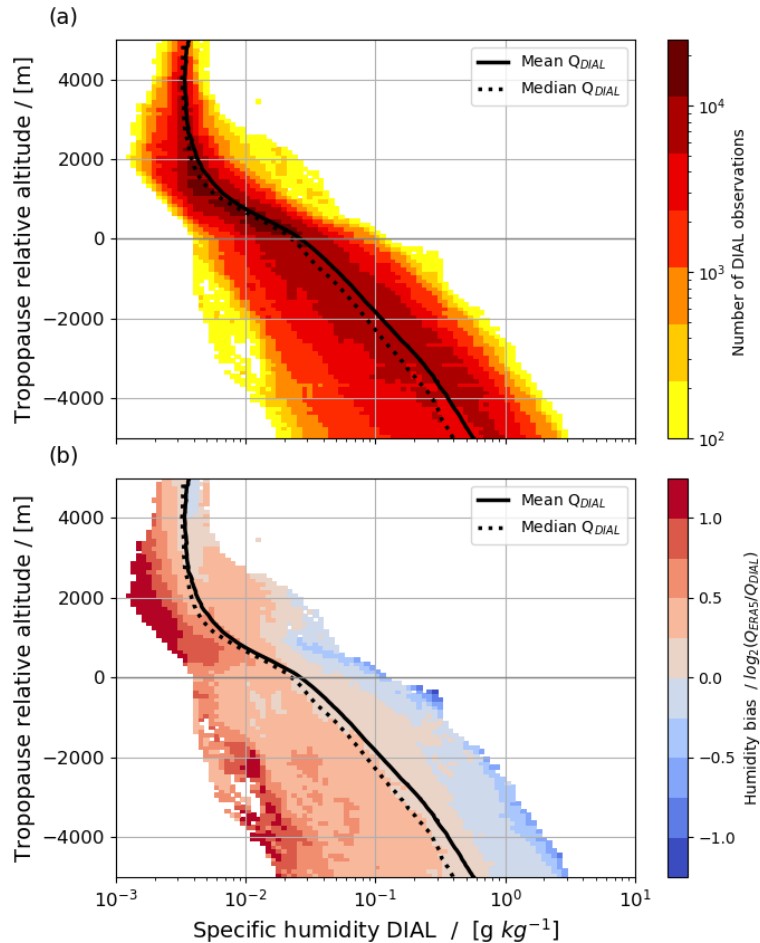

**Figure 6:** Binned distribution of DIAL specific humidity observations relative to the thermal tropopause coloured by (a) the number of observations per bin and (b) the bin-average humidity bias. Thick black solid (dotted) lines in (a) and (b) show mean (median) values per altitude bin. Bin sizes are 100 m and 0.01 g kg$^{-1}$ of log$_{10}$ (q$_{DIAL}$). Please note the logarithmic abscissa and colour bar in (a).

### 3.2.2 Synoptic and seasonal variability

In this section, the variability of the LS bias between flights, campaigns and tropopause altitudes is investigated. Figure 7 shows the observed humidity distribution within a 3–km layer above the tTP, i.e., the area of the strongest LS moist bias. The observed humidity values of all flights range from $1 \times 10^{-2}$ to $4 \times 10^{-3}$ g kg$^{-1}$ and their interquartile range strongly varies between the individual flights, which presumably relates to differences in the flight level, the tropopause altitude and the synoptic situation. During summer (CIRRUS-HL) and fall campaigns (NAWDEX, WISE) the range of observed humidity is larger compared to the winter campaign (NARVAL). It is furthermore noticeable that intra-campaign variations (i.e., synoptic variability) of observed humidity exceed the seasonal variability. Per flight, the median LS bias (Fig. 7b) varies from 0.2 (15 %) to 1.4 (164 %), but a positive bias is detected for each flight. Whereas the magnitude of the bias shows no obvious correlation

with the LS moisture distribution, the moist bias appears to be smaller in winter (NARVAL) compared to fall (NAWDEX,
WISE). Interestingly, the moist bias during the CIRRUS-HL summer campaign is remarkably strong. The number of
observations that is available for each flight is strongly variable between a few and several hundred thousand (Fig. 7c).

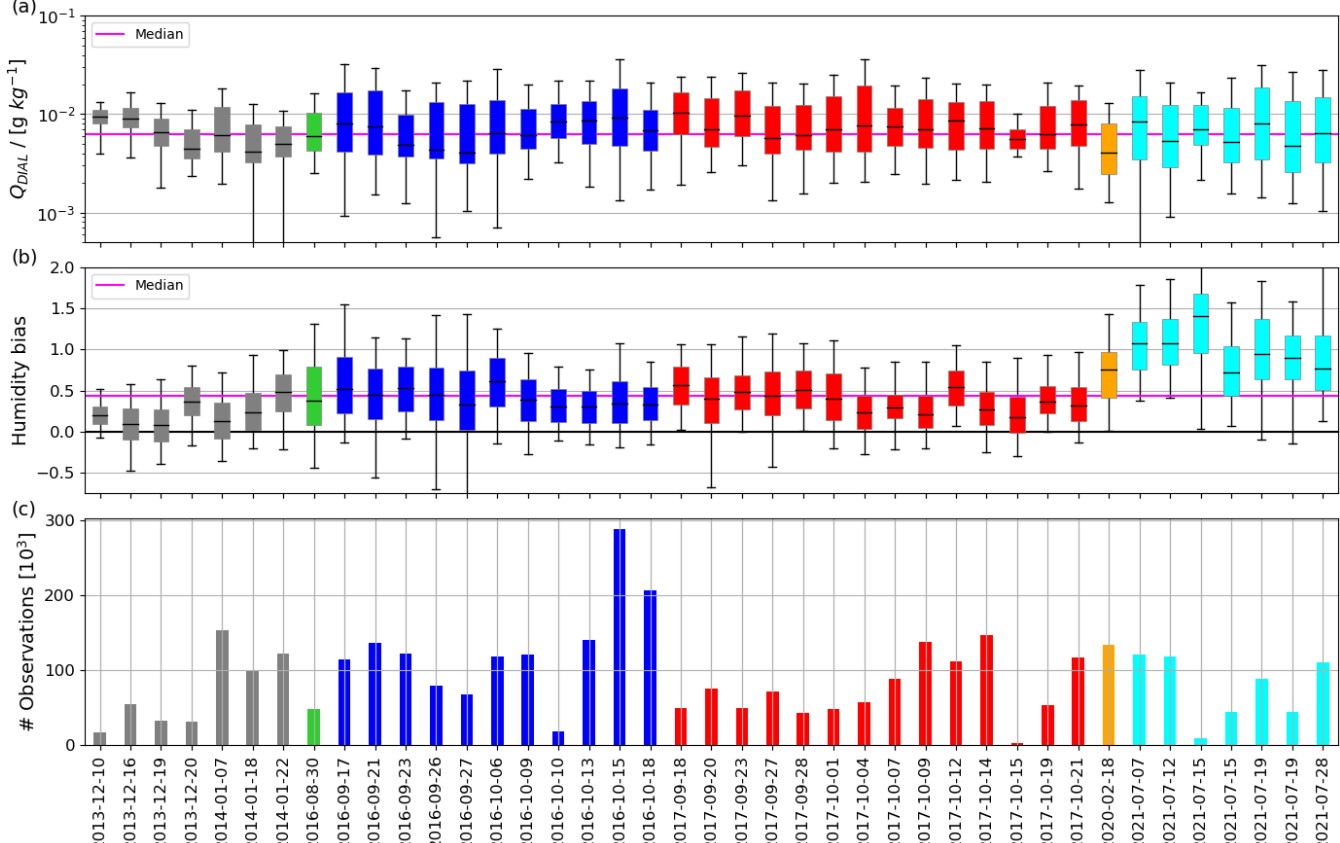

**Figure 7:** Distributions of (a) the observed humidity and (b) the humidity bias and (c) the number of data points for the layer 0 to 3 km
above the tTP. The average observed humidity and bias for all flights is given by the magenta lines in (a) and (b). The boxes in (a) and (b)
define the interquartile range located around the median (black) and the whiskers illustrate the 5th/95th percentile. The different campaigns
are colour-coded as in Fig. 1.

The average profile of the bias and the number of observations for campaigns with an increased data coverage is shown in Fig.
8. The data availability is very different across the campaigns (Fig. 8b). During NAWDEX and WISE a large number of
observations is present between -5 km below and 4 km above the tTP. CIRRUS-HL provides approximately half as much data
at each altitude except for altitudes beyond 3 km above the tTP where little data is available. Due to frequent low flight levels
during the seven NARVAL flights, only a small number of observations is available beyond 1 km above the tTP. The general
structure with a pronounced positive bias, a local minimum at the tropopause and a decrease towards the highest altitudes is
apparent for all campaigns (Fig. 8a), although significant differences can be identified across the campaigns. For the fall

campaigns in two successive years (NAWDEX and WISE) a similar shape of the bias is observed across the entire profile. The
345 maximum moist bias is located at approximately the same altitude, and a similar decrease beyond this maximum is observed.
However, the magnitude of the LS moist bias is slightly higher for NAWDEX (0.6, 50 %) compared to WISE (0.5, 40 %).
During summer (CIRRUS-HL), a stronger moist bias is detected exceeding 1.2 (130 %) at its maximum. Compared to fall, the
summer bias is increased by a factor of 2–3. During winter (NARVAL), the LS moist bias is small (0.3, ~23 %) and not
substantially higher than the upper-tropospheric bias, but the limited representativity due to the low number of observations
should be noted here.

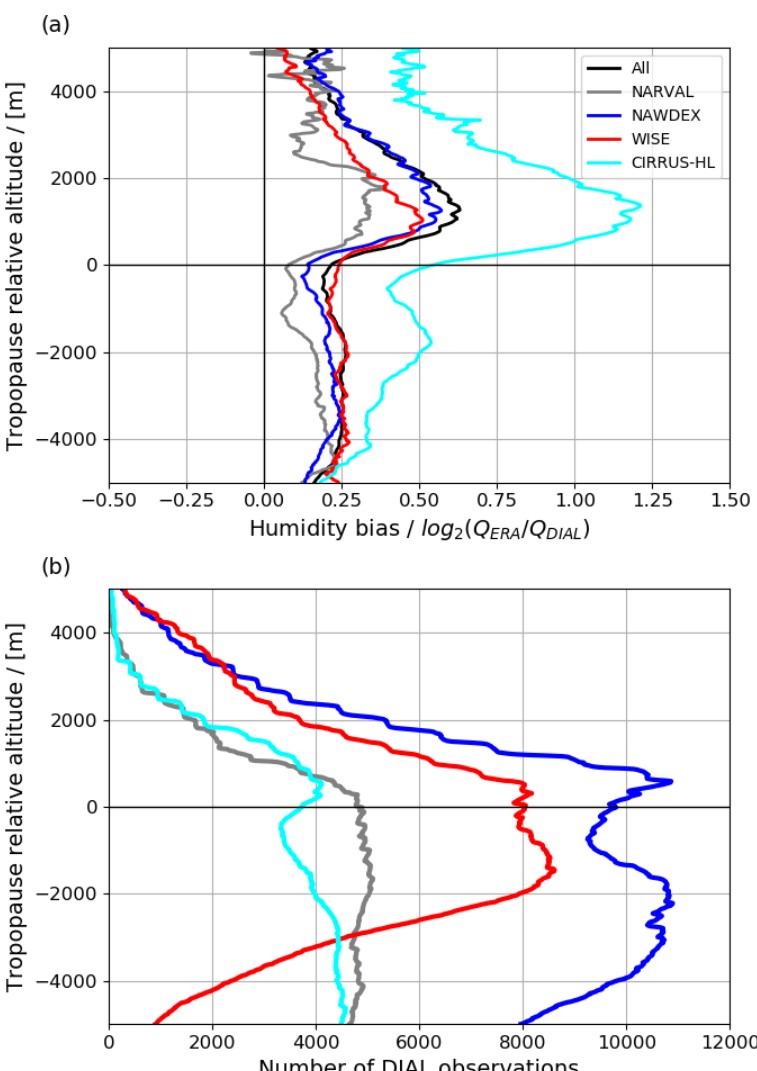

**Figure 8:** Tropopause-relative vertical profiles of (a) humidity bias and (b) number of observations for the different campaigns (colour-
coded as in Fig. 1). The black line represents the multi-campaign average.

In addition, we explore whether the observed vertical structure of the moist bias is sensitive to different synoptic situations.

For this investigation, the DIAL profiles are classified by their corresponding tTP altitude. Lower tropopauses are typically associated with trough situations and high tropopauses occur above ridges. For each category the corresponding average bias profile and the number of observations is given in Fig. 9. The vertical structure of the bias (Fig. 5b) is reproduced for each tropopause altitude interval. No systematic differences between the bias profiles can be revealed in the UT. Interestingly, each category shows an increased moist bias of comparable magnitude as well as a decrease above, although its vertical position

relative to the tTP is different. The maximum bias is located higher for low tropopause altitudes, while profiles with high tTP altitude show a maximum closer to the tTP. For instance, the maximum bias for low tropopauses ($< 8$ km) is located at 2 km above the tTP, while for the category with highest tropopauses (12–14 km) the maximum value is found at 1 km. The number of data points illustrates that each category exhibits a reasonable number of observations (Fig. 9b).

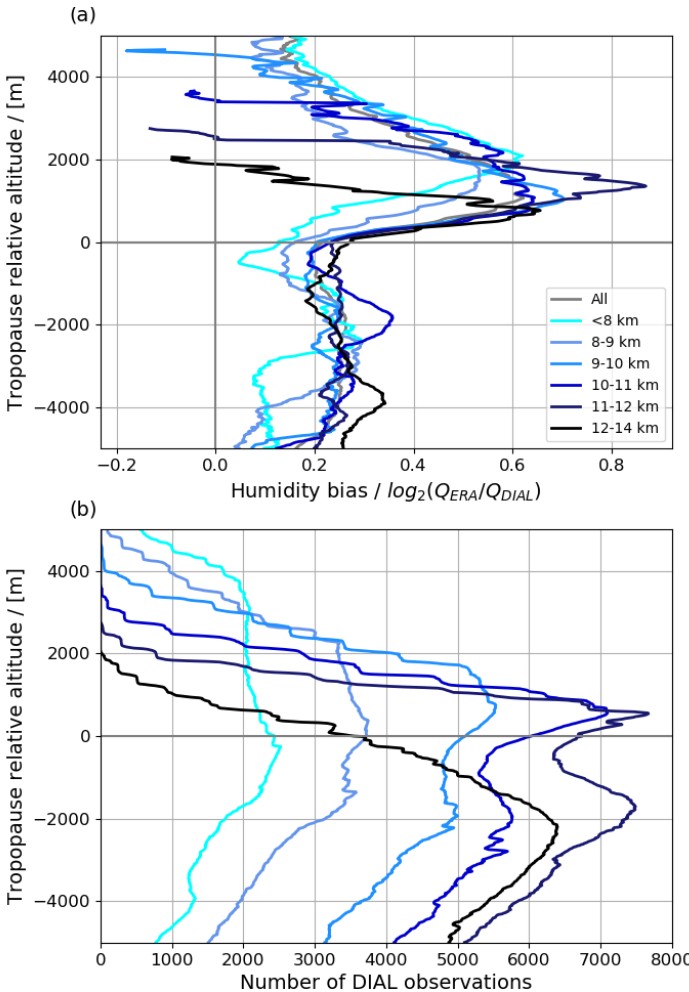

**Figure 9:** Tropopause-relative profiles of the (a) humidity bias and (b) number of observations for different intervals of tTP altitudes (colour-coded).

### 3.3 The vertical structure of the moist bias related to mixing processes

In the following it is examined to what extent the observed air masses have experienced mixing in their history and whether this is related to the vertical structure of the moist bias. For this purpose, we examine collocated ozone and water vapour observations that were collected during four WISE research flights and that provide a suitable data coverage. First, the observed ozone distribution for the same case study as introduced in section 3.1 is shown in Fig. 10a. Note that the ozone and water vapour observations are given as volume mixing ratios (VMR) in the following. The distribution of ozone is opposite to that of water vapour, with lowest concentrations ($VMR_{O3} < 100$ ppb) in the troposphere and an increase with altitude across the tropopause to $VMR_{O3} > 500$ ppb. Note the filamentary structures of increased ozone values in the LS and the ozone-rich air which is transported downward within the tropopause fold (see detailed description in Schäfler et al., 2022).

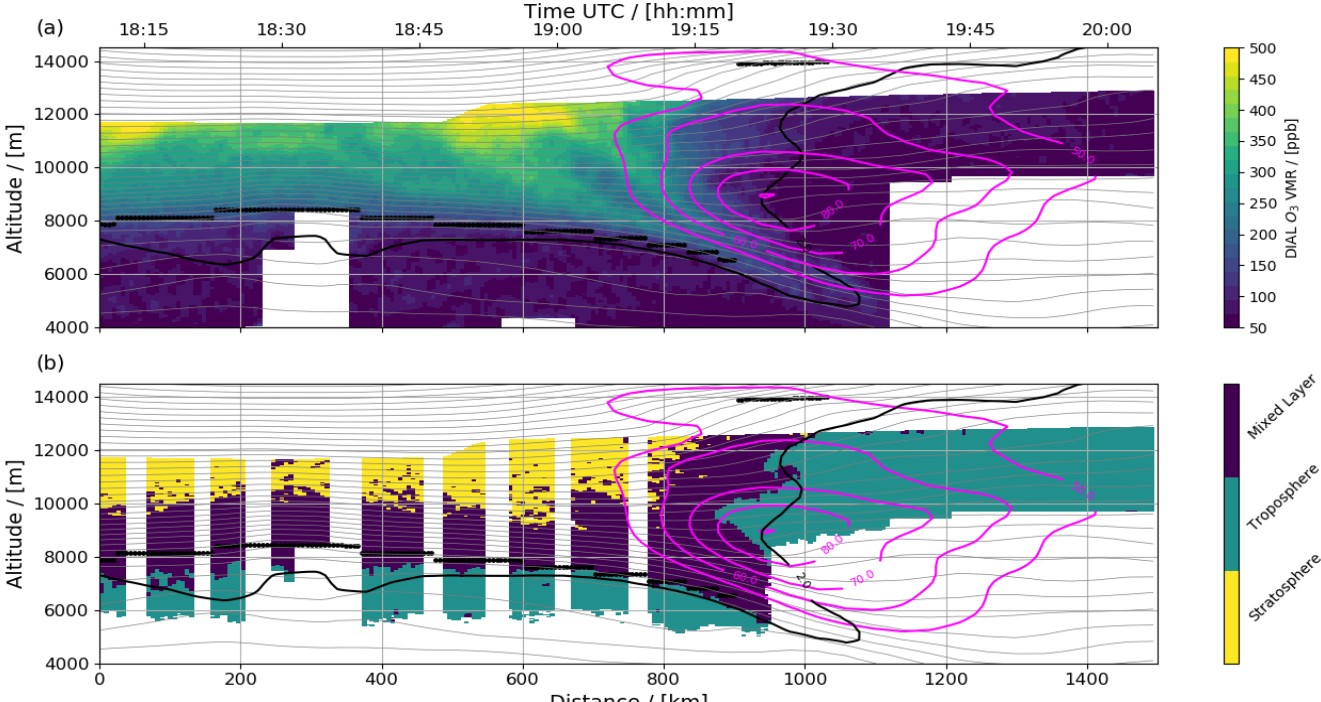

**Figure 10:** Vertical cross sections as in Fig. 4, but for (a) DIAL ozone volume mixing ratio (in ppb) and (b) air mass classes derived from water vapour and ozone measurements (for details see Sect. 3.3).

Following the approach by Schäfler et al. (2021), the collocated water vapour and ozone observations for four WISE flights are illustrated in tracer–tracer (T–T) phase space in Fig. 11 and three classes of observations are identified based on the characteristic distributions (e.g., Pan et al., 2004). First, tropospheric observations are characterized by low $VMR_{O3}$ (typically < 100 ppb) and a large spread of $VMR_{H2O}$. Second, high $VMR_{O3}$ at low $VMR_{H2O}$ (< 6.5 ppm or < 4x10$^{-3}$ g kg$^{-1}$) are assigned to lower stratospheric air. Additionally, a class with intermediate chemical characteristics ($VMR_{H2O} > 6.5$ ppm and

VMR$_{O3}$ > 100 ppb) is attributed to mixed air masses that experienced mixing between the troposphere and stratosphere.
Although Schäfler et al. (2021) suggested a careful selection of the threshold for individual flights, here constant values (see caption of Fig. 11) are used for all four WISE flights, which is sufficiently accurate for our objective. Sensitivity tests with slightly varied thresholds have shown only a little impact on the distribution of the classes in geometrical space. Such a re-projection of the air mass classification from T–T space to geometrical space with a coherent distribution of the three classes is shown in Fig. 10b. Observations below the tTP are predominantly assigned to tropospheric air, while the uppermost data to the north of the jet stream is classified as stratospheric air. South of the jet stream, where the flight altitude is below the tTP, only tropospheric air is detected. In between the tropospheric and the stratospheric air, the mixed air mass is following the tropopause in a 2–3 km thick layer, which appears to be vertically deeper in the tropopause fold.

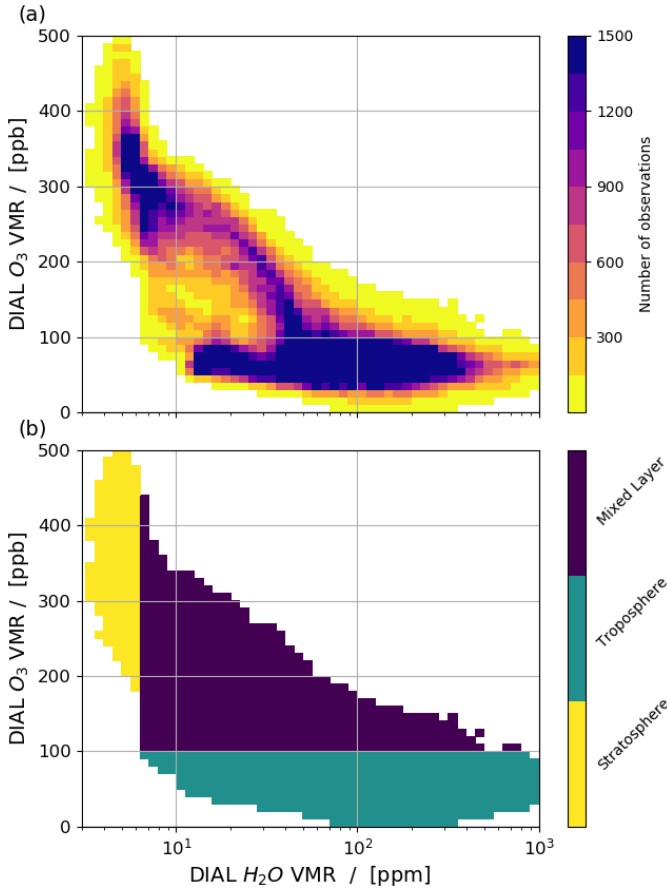

**Figure 11:** Binned distribution of water vapour and ozone observations in T–T space for four WISE flights coloured by bin-average (a) number of DIAL observations and (b) type of classified air mass with troposphere (VMR$_{O3}$ < 100 ppb and VMR$_{H2O}$ > 6.5 ppm), mixed air (VMR$_{O3}$ > 100 ppb and VMR$_{H2O}$ > 6.5 ppm) and stratosphere (> 100 ppb VMR$_{O3}$ and < 6.5 ppm VMR$_{H2O}$). Bin sizes are 10 ppb for VMR$_{O3}$ and 0.05 ppm for $\log_{10}$ (VMR$_{H2O}$).

For each bin in T–T space the average humidity bias and the mean tropopause-relative altitude are displayed in Fig. 12. The humidity bias is weak for both tropospheric and stratospheric air (Fig. 12a and Fig. 11b) ranging mostly between -0.25 and 0.25. In the mixed air class, the humidity bias is most pronounced (> 0.25), particularly where the $VMR_{H2O}$ is below 40 ppm. In the tropospheric and stratospheric air, a stronger positive/negative bias is indicated for lower/higher $VMR_{H2O}$ which is associated with the sharper humidity distribution in ERA5 (see discussion in Sect. 3.2.1). Figure 12b displays the tropopause-relative height, which is the vertical distance to the tTP, for each bin. Across the mixed air class an increase of the tropopause-relative altitude is visible corresponding to a decrease of $VMR_{H2O}$ and to an increase of $VMR_{O3}$. At low $VMR_{H2O}$ (< 10 ppm) and low $VMR_{O3}$ (100–200 ppb) the transition of tropopause-relative altitudes is more abrupt which is related to tTP variability across the jet stream, e.g., as visible in the uppermost part of Fig. 10b. When comparing the tropopause-relative height with the distribution of the bias (Fig. 12a) it is noticeable that the average bias is particularly increased between 1 and 3 km where it ranges from 0.5 (40 %) up to 1.25 (137 %). In contrast, the mean bias is weak beyond 3 km above the tTP and below the tropopause.

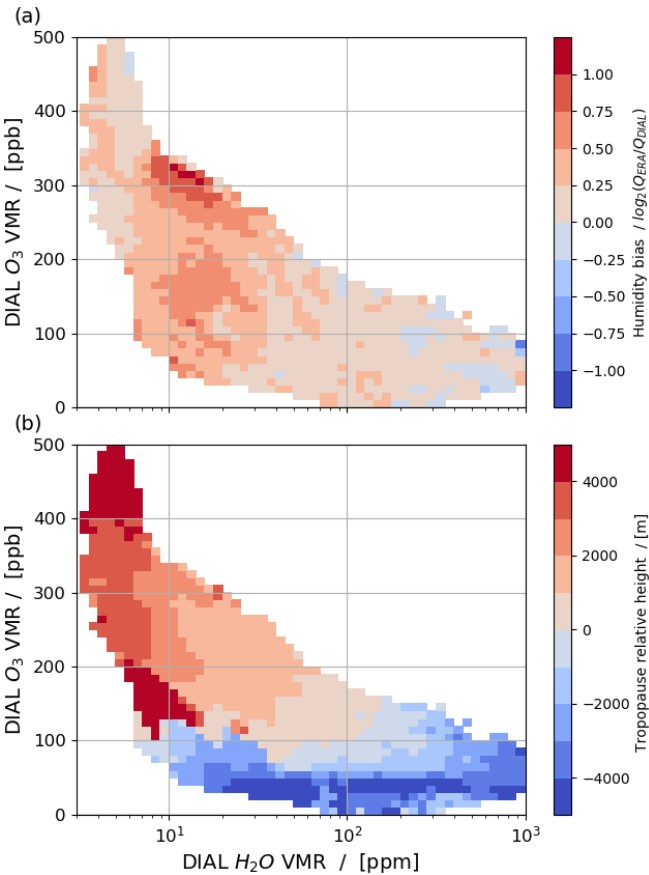

**Figure 12:** Binned distribution of water vapour and ozone observations in T–T space as in Fig. 11 but coloured by bin-average (a) humidity bias and (b) tropopause-relative altitude.

The average vertical profile of the moist bias for the WISE flights (Fig. 13a) is similar to the full dataset (Fig. 5b) at the tTP
and in the LS, i.e., a local minimum is found at the tTP (0.1; 7 %) and a pronounced maximum of 0.62 (54 %) peaking at about
1 km above the tTP. The tropospheric part of the profile, however, is almost constant in the full dataset (0.2–0.25) but
decreasing with increasing altitude in the WISE data (0.4–0.1). Figure 13b shows the relative proportion of the individual air
masses at a given tropopause-relative altitude and thus gives information about the connection between the vertical structure
of the moist bias and the air mass classes. In the entire UT, the tropospheric air provides the largest contribution of more than
80 % up to 500 m below the tTP. Across the tTP, the proportion of tropospheric air rapidly decreases with altitude in accordance
with a rapid growth of the fraction of mixed air. This is accompanied by an increase of the moist bias and the altitude of the
largest bias (1–2 km above the tTP) coincides with the maximum relative contribution of the mixed air class (> 90 %). Above,
the relative fraction of stratospheric air grows, while the moist bias reduces and reaches constant values (0.2) at ~4 km above
the tTP with a 65–85 % share of stratospheric air. Please note that contributions of mixed air below the tropopause and at
altitudes > 4 km above the tTP may be related to falsely detected tropopause altitudes (see discussion in Sect. 4) or situations
of complex tropopause structure (e.g., as shown in the second part of Fig. 10b).

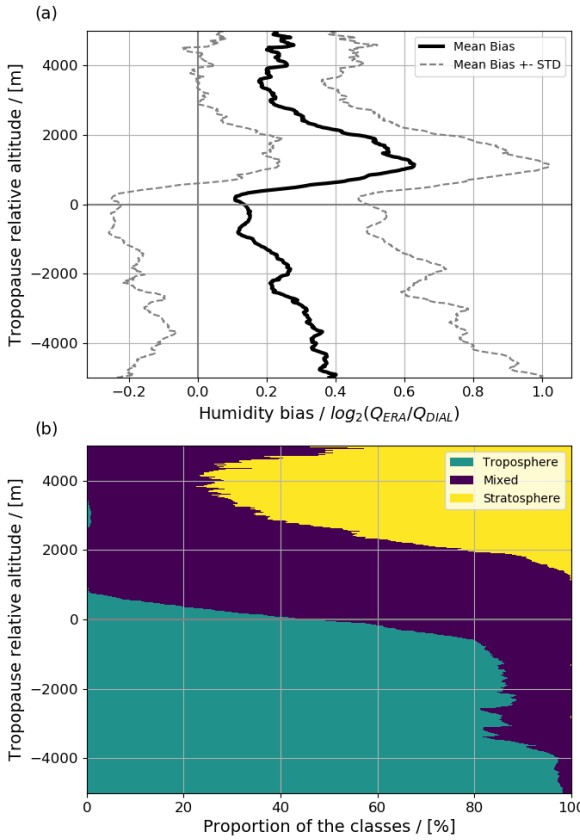

**Figure 13**: (a) Tropopause-relative vertical profile of the mean (thick black line) and standard deviation (thin grey dotted lines) of the
humidity bias and (b) relative proportion of the individual air mass classes for four WISE flights.

## 4 Discussion

Recent studies document a lower-stratospheric moist bias in different NWP models (e.g., Kunz et al., 2014; Dyroff et al., 2015; Kaufmann et al., 2018; Woiwode et al., 2020; Bland et al., 2021). We find a comparable moist bias in ERA5 reanalyses based on a comprehensive multi-campaign water vapour lidar data set comprising 41 research flights (6 campaigns) and roughly 33000 vertical profiles obtained in the northern hemispheric midlatitudes during different seasons. The observations from the surface up to the LS cover four orders of magnitude and represent typical midlatitude data for the individual seasons (e.g., Pan et al., 2000; Randel and Wu, 2010, Zahn et al., 2014; Kunz et al., 2014; Dyroff et al., 2015; Bland et al., 2021). The high data availability around the tropopause makes the data set suitable for an evaluation of NWP fields in the UTLS. Although the number of observations reduces considerably towards the highest altitudes (up to 5 km above the tTP), the data set provides a valuable extension to previous humidity data sets which exhibit increased measurement uncertainties at altitudes larger than 2 km above the tTP (e.g., Bland et al., 2021).

In the troposphere we find strong positive and negative biases of small spatial extent, which are likely related to insufficiently represented tropospheric transport processes, to model errors of tropospheric processes (e.g., clouds) or to the linear interpolation scheme that may have caused increased differences especially in situations of strong horizontal or vertical moisture gradients. The small positive and vertically almost constant mean bias in the UT, which ranges between 0.2 (15 %) and 0.26 (20 %), confirms earlier findings (Dyroff et al., 2015; Bland et al., 2021). It has to be noted that the UT bias is limited to cloud-free scenes, as DIAL humidity profile observations cannot be retrieved inside or below optical thick clouds. In agreement with Bland et al. (2021) a local minimum of the bias ($< 0.2$, $< 15$ %) is found at the tTP. Above the tropopause, our findings confirm a coherent layer of overestimated humidity in ERA5 reanalyses. The magnitude of the maximum bias of 0.63 (55 %) and its altitude of 1.3 km above the tTP is comparable to previous findings for earlier model cycles of the IFS (Dyroff et al., 2015; Kaufmann et al., 2018; Woiwode et al., 2020; Bland et al., 2021), earlier reanalysis versions (Oikonomou and O'Neill, 2006; Kunz et al., 2014) and other evaluated models (Davis et al., 2017; Bland et al., 2021). Above the maximum bias, in a region where recent studies present diverging results (Dyroff et al., 2015; Woiwode et al., 2020; Bland et al., 2021) our analysis reveals a steadily decreasing moist bias that reduces to nearly constant and small positive values comparable to the UT. The independence of measurement error from altitude and humidity concentration allows a reliable and robust depiction of the bias at the highest altitudes of the UTLS. Furthermore, the magnitude of the LS moist bias exceeds the expected error of the DIAL humidity observations by approx. one order of magnitude which underlines the significance of our results. Please note that Bland et al. (2021) show that tTP altitudes are on average about 200 m higher when derived from ECMWF IFS profiles compared to radiosondes which may impact tropopause-relative moisture distributions and in turn the bias. As no temperature observations are available, this study relies only on ERA5 tTP altitudes. Assuming a systematic shift by 200 m would reduce the tropospheric bias, however, the LS moist bias, although slightly weakened would persist.

In line with findings of Bland et al. (2021) who indicated little sensitivity of the moist bias to various atmospheric conditions but revealed a different depth of the moist bias for trough and ridge situations, low tTP situations (which are typically associated

with troughs) exhibit a maximum bias at higher altitudes and a deeper layer of the increased bias compared to high tTP situations. The magnitude of the moist bias is found to be independent of the tropopause altitude. In addition, we detect a pronounced LS moist bias in the summer (> 1.20, > 130 %) which exceeds the diagnosed autumn bias by a factor of 2–3. So far, such a seasonality was only suggested in Dyroff et al. (2015). Additional DIAL observations in spring, summer and winter would be valuable for a more comprehensive study of the seasonality of the vertical bias structure.

For four flights during the WISE campaign an air mass classification using collocated water vapour and ozone profile data (Schäfler et al., 2021) was applied to separate tropospheric (low ozone and large water vapour mixing ratio), stratospheric (large ozone and low water vapour) and mixed air (intermediate ozone and water vapour). In tropopause-relative coordinates, the vertical structure of the moist bias for the selected cases turned out to be comparable to the multi-campaign LS moist bias, so that these flights are considered to be representative for autumn. We find that the moist bias is increased in the mixed air class representing the ExTL and that the maximum is reached at the altitude where the proportion of mixed air is highest (near 100 %). The decrease of the moist bias above/below is accompanied by a growth of the proportion of stratospheric/tropospheric air. The high correlation in the distribution of the moist bias and the ExTL gives a strong hint at the importance of moisture injection into the LS, either due to numerical diffusion across the tropopause or due to insufficiently modelled transport and mixing processes. As the bias in the ExTL is increased in each of the evaluated WISE flights we consider systematic uncertainties in the representation of mixing processes to play a key role for the LS moist bias. This is supported by the finding of a deeper bias layer above troughs which are characterized by a thicker ExTL above (e.g., Hoor et al., 2002; Pan et al., 2007). In addition, the maximum bias occurs in summer when cross-tropopause mixing is strongest (Hoor et al., 2002) and, finally, the bias is reduced in stratospheric background humidity at highest altitudes, which are not influenced by mixing processes at the extratropical tropopause. Schäfler et al. (2022) investigate the Lagrangian history of the observed air for the presented WISE case study on 1 October 2017 and find that the ExTL air experienced strong turbulent mixing in the jet stream during 48 h before the observation. They also find that the mixed air (in which we identified the increased bias) shows highly variable origins and transport pathways related to tropospheric weather systems which may be indicative for the relevance of different mixing processes. Additional collocated observations of ozone and water vapour in different seasons, near active mixing process (e.g., convection) or in the southern hemisphere where exchange at the polar jet stream is reduced (e.g., Bowman, 1995) could provide valuable information about the relevance of individual mixing processes and their role in forming the moist bias. The presented results suggest that improving the representation of mixing at the tropopause may reduce the humidity bias and be beneficial to improve the modelling of climate and weather. Davis et al. (2017) demonstrates that various reanalyses significantly overestimate LS humidity in the extratropics. The systematic moist bias in ERA5 reanalyses has to be kept in mind for climatological studies using ERA5 humidity fields in the LS.

**5 Conclusion**

In this study we applied a comprehensive data set of airborne water vapour lidar profiles to investigate the representation of
specific humidity in the ERA5 reanalysis across the extratropical UTLS. The main conclusions of this work are summarized
below following the three research questions that were raised in the introduction:

1. Can the multi-campaign DIAL data set robustly quantify the LS moisture bias in ERA5?

The presented DIAL data set with its a large number of high-accuracy and high-resolution humidity profiles measured over
the North Atlantic and Europe during six research campaigns between 2013–2021 provides a valuable extension to the
available observational data sets that were used to determine the lower-stratospheric moist bias. Beside the broad range of
observed humidity values ($10^{-3}$ to $10^1$ g kg$^{-1}$), especially the high data availability in the ±5 km around the tropopause makes
the data suitable for the characterization of water vapour in the entire midlatitude UTLS. The flights that were performed in
different times of the year indicate seasonal differences in the observed humidity distributions. As the flights also cover diverse
synoptic situations we consider the data set to be representative for the midlatitudes. The data set holds the advantage of not
being assimilated by NWP and thus allows humidity errors in the ERA5 reanalysis to be evaluated independently.

2. *What is the vertical structure of the LS moist bias in ERA5, particularly at high altitudes?*

Our analysis demonstrates that a systematic lower-stratospheric moist bias is also present in ECMWF's most recent global
reanalysis ERA5. We find that the vertical structure of the bias, that is analysed in tropopause-relative coordinates, is
characterized by a weak positive bias in the upper troposphere (15–20 %), a strong overestimation of humidity that reaches a
maximum (55 %) at 1.3 km above the thermal tropopause. Above this maximum, we detect a steady vertical decrease of the
moist bias towards a constant small value (15 %) beyond 4 km above the tropopause. The moist bias occurs in coherent and
extended regions along the individual lidar cross-sections. The above described unique measurement characteristics of the
DIAL data set together with the persistence of the bias structure in different flights and campaigns allow the vertical decline
at the highest altitudes to be robustly confirmed. A high similarity for two campaigns conducted in the same region over the
North Atlantic in successive years illustrates the persistence of the vertical structure. We find a seasonality of the moist bias
with a maximum in summer and a minimum in winter. Lower tropopause altitudes, which are typically related to troughs,
exhibit a deeper layer of increased moist bias while the moist bias over ridges is confined to a shallow layer.

3. *Is the moist bias correlated to the distribution of mixed air masses in the UTLS?*

For four flights of the DIAL data set collocated water vapour and ozone profiles are available and used to classify UTLS air
masses according to their chemical characteristics into tropospheric, stratospheric and mixed air. We find the strongest bias at
altitudes dominated by the mixed air class representing the ExTL while tropospheric or stratospheric air exhibit a smaller bias.
From this correlation we deduce that insufficiently represented mixing processes and the role of numerical diffusion in ERA5
shape the vertical structure of the lower-stratospheric bias with the maximum occurring at altitudes that are most frequently
affected by exchange processes between the troposphere and the stratosphere. The vertical structure of the moist bias of the

entire data set is comparable to the four flights with collocated ozone and water vapour observations. In addition, the deeper bias over troughs which typically feature a deeper ExTL, the maximum moist bias in summer when cross-tropopause mixing is strongest, and the reduced bias at altitudes of constant stratospheric background humidity leads to the conclusion that the findings are applicable to the midlatitude in general. In the future, it would be interesting to identify the individual mixing processes that affect the moist bias most and the time scales on which it is formed.


**Data availability**

The lidar data used in this study are available through the HALO database (https://halo-db.pa.op.dlr.de/). We are grateful to ECMWF for granting access to the full-resolution ERA5 data.

**Author contributions.**

KK performed the data analysis, produced the figures and wrote the manuscript. AS, MWi, MWe and GC supported the interpretation of the data, contributed with ideas and commented on the paper. MWi performed the DIAL data processing.

**Competing interests**

The authors declare that they have no conflicts of interest.

**Acknowledgements**

The authors thank the individual research teams that successfully conducted the field campaigns NARVAL, NARVAL2, NAWDEX, WISE, EUREC[4]A and CIRRUS-HL which enabled us to perform this study. This work was supported by the Transregional Collaborative Research Center SFB/TRR165 "Waves to Weather" (https://www.wavestoweather.de) funded by the German Research Foundation (DFG). We further acknowledge the DFG for supporting the HALO missions within the priority program SPP 1294 "Atmospheric and Earth System Research with HALO" (https://www.halo-spp.de/). We are

grateful to DLR who supported this work in the framework of the DLR project "Klimarelevanz von atmosphärischen Spurengasen, Aerosolen und Wolken" (KliSAW). We thank Andreas Dörnbrack for his valuable comments on the manuscript.

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
