# Peer review of "Vertical structure of the lower-stratospheric moist bias in the ERA5 reanalysis and its connection to mixing processes"

_Atmospheric Chemistry and Physics, 2022_

## Author Comment (AC1)

**ACP-2022-505 – Vertical structure of the lower-stratospheric bias in the ERA5 reanalysis and its relation to mixing processes**

By Krüger et al. (2022)

Reply to the Editor

I agree with the referees that this is in excellent study. While waiting for the second referees comment I also had a look at the manuscript and have a bunch of rather technical comments which I would like the authors to consider when they do the revision. *We thank the editor for the positive feedback and the careful revision of our manuscript. We answered each comment below using a blue font.*

**Generally comment:**

I would suggest to write throughout the manuscript "moist bias" instead of just "bias" and "jet stream" instead of just "jet". *We followed the editor's recommendation wherever applicable.*

**Specific comments:**

P1, L10: What kind of errors/uncertainties does this cause? Regarding e.g weather, does it affect predictions for cloud formation and/or precipitation? Can you give an example? *We do not want to detail the impact on climate and weather predictions in the abstract, but the reader can find all relevant information in the introduction and discussion sections. Future studies may focus on implications of the bias and ways to improve the analyses. These points are partly raised in the discussion.*

P1, L13: Mention here also how many campaigns were preformed. *Corrected. We included the number of campaigns.*

P1, L14: The number of 31000 humidity profiles is impressive, However, could you also provide the time resolution of the data?

*We think that this is too much detail for the abstract, but of course the reader can find this information about temporal and spatial resolution of the data in Sec. 2.1.*

P1, L21: Add here "(O3)" after ozone and "(H2O)" after water vapour to introduce the abbreviations that are used then in the next line. *Thank you for pointing to this inconsistency. Instead of using the abbreviations we consistently rely on "water vapor" and "ozone" throughout the manuscript.*

P1, L26: I would add after positive impact "i.e. more accurate" since I think that will be the outcome positive impact of a lower moist bias on the forecasts. *We clarified this sentence and replaced the "positive impact on weather and climate forecasts" by "lead to more accurate weather and climate forecasts".*

P1, L20: See my comment on P1, L13. That is why I suggest to mention above already that several campaigns have been performed. Otherwise writing here "During one campaign" is rather confusing. *We included the number of campaigns as documented above and therefore think "during one campaign" should be clear now.*

P2, L29ff: This paragraph is rather confusing. I would suggest to revise it. Start with describing the general vertical distribution of H2O (high in in the troposphere, low in the stratosphere) and then discuss the gradient and the exchange between troposphere and stratosphere. The vertical distribution of H2O is not only dynamically driven, also the chemical sources and sinks of H2O play a role.

*We cannot completely understand where the confusion comes from. The first sentence highlights the general relevance of water vapor. We removed the second sentence (impact on the T-profile), which is repeated later and might be confusing. Then we actually start with the distribution and its UTLS*

impact, which is at the centre of this study. Afterwards, we close the paragraph with relevance of the vertical structure for weather and climate.

We agree that the dynamical processes are only part of the complete story. Therefore, we revised a sentence in the second paragraph, which now reads as follows: *"In the extratropical UTLS, the distribution of water vapor is driven by transport and mixing processes related to baroclinic waves and associated synoptic and meso-scale weather systems, which are interacting with chemical processes (e.g., Gettelmann et al., 2011; Schäfler et al., 2022)."*

P2, L56: co-loacted cold bias. Where exactly is this bias located? Also in the stratosphere? Or in the atmospheric layer below (troposphere) or above (mesosphere)? Updated. In order to be more specific, we added "*at the same altitudes*" to the sentence.

P3, L65: "for" is not correct here. Do you mean "from" different radiosonde types (thus two types of radiosondes show the opposing vertical structure) or do you mean "by comparing to two different radiosonde types"? In the updated version, we now use *"from"* instead of *"for"*.

P3, L65ff: I would suggest to rephrase the sentence and to start with what Woiwode et al. compare with and then what the result is. Corrected as suggested.

P3, L71ff: This sentence/paragraph is also rather confusing and should be improved. Numerical diffusion is something that affects Euler models, especially if a rather course grid is used, thus naming in this sentence the "semi-Lagrangian" is a bit confusing. Further, two my knowledge not the ECMWF model itself is Lagrangian, its rather one of the schemes used in the model that is semi-Lagrangian. You actually name it, it is the advection scheme. So it should read rather "the semi-Lagrangian advection scheme used in the ECMWF model". Corrected as suggested.

P3, L79: highest altitude → give an approximate altitude. Instead of *"highest altitude"*, we use *"> 2 km above the tropopause"* in accordance with Bland et al. (2021)

P3, L82: The reference of Hegglin et al. (2009) is not correct here. In that study ACE data has been used which is a solar occultation instrument and not a microwave sounder. I think the Hegglin reference you meant here is the paper where the H2O climatologies are compared (Hegglin et al., 2013, JGR). Also here additionally some of the WAVAS-II comparison papers should be cited (e.g. Lossow et al., (2017, 2018), Khosrawi et al. (2018), Read et al. (2022), see WAVAS special issue https://acp.copernicus.org/articles/special_issue830.html although I am not sure if one of the studies explicitly mentions the problem with the vertical resolution of the microwave sounders. We are grateful for pointing to the incorrect reference and for suggesting Khosrawi et al. (2018), which we were not aware of. The references were updated.

P4, L101: Not clear what you mean here with air mass classes. Through removing the term "*air mass class*", we tried to clarify the sentence in the outline, which now reads as follows: *"The relationship between the vertical structure of the moist bias and the distribution of tropospheric, stratospheric and mixed air is presented in Sect. 3."*

P4, L113: Rephrase. Not the wavelengths itself consist of online channels. Rather the observations at these wavelengths are separated into the/by the channels". Corrected as suggested.

P4, L116: The wavelength ranges have been optimized? We now use "*are selected*" instead of "*have been optimized*".

P4, L116: Also here rephrase the sentences. Generally, the whole paragraph should be revised. We considered the two previous comments to make this paragraph clearer. Apart from this, we kept this paragraph, as it was not clear to us what needs to be changed to further improve comprehensibility.

P5, L155: This sentence is also rather confusing. Better to write "parts of the HALO flight tracks of all research flights where DIAL observations where obtained." Corrected as suggested. We furthermore

revised the whole paragraph (starting P5, L137 in the preprint version) to give it a clearer structure. This paragraph now reads as follows:

*"In this study, we use DIAL observations from six campaigns from 2013–2021 that provide almost 33000 water vapour profiles obtained during 41 research flights. The profiles were sampled along the flight track and extend from the surface up to about 14 km altitude corresponding to the maximum flight level of the HALO aircraft (Krautstrunk and Giez, 2012). As the focus of this study is the midlatitude UTLS, we only consider flights that provide a significant amount of data across the tropopause. The majority (25) of these flights took place in the northern hemispheric fall season during the North Atlantic Waveguide Downstream impact EXperiment (NAWDEX; Schäfler et al., 2018) and the Wave-driven ISentropic Exchange campaign (WISE; Kunkel et al., 2019). As part of the campaigns ElUcidating the RolE of Cloud-Circulation Coupling in ClimAte (EUREC⁴A; Stevens et al., 2021), the Next-generation Aircraft Remote sensing for VALidation studies (NARVAL; Klepp et al., 2014) and NARVAL2 (Stevens et al., 2019) measurements were taken during eight flights in winter season. In addition, the Cirrus in High-Latitudes (CIRRUS-HL) mission provides observations in summer. Figure 1 depicts the parts of HALO research flights where DIAL observations were obtained. Most flights were carried out over the North Atlantic between 48 °N and 66 °N, the North Sea and central to western Europe. Additionally, the subtropics (> 35 °N) and the Arctic were covered by individual flights as well.*

*During the WISE campaign, WALES was operated in a different setup to measure both water vapour and ozone, concurrently. For this purpose, two of the 935 nm NIR water vapour channels were replaced by two ultraviolet (UV) channels covering the 300–305 nm ozone absorption line (Fix et al., 2019). The use of two instead of four channels per trace gas leads to a reduced vertical coverage which was optimized so that the selected NIR wavelengths cover the tropopause region. Increased statistical noise required averaging over a period of 24 s (~6 km horizontally) while the effective vertical resolution remains approximately 300 m (Fix et al., 2019)."*

P8, L186: T639? Is there something missing? Usually the T gives the horizontal resolution and the L the number of levels. We understand this comment as one can find different terminologies in literature. However, we follow table 2 in Hersbach et al. (2020) where "TL639" to used for the spatial resolution of ERA5.

P8, L192: You mean you do here the conversion from sigma coordinates to pressure coordinates? We don't want to go into more detail here as the procedure is documented in the given ECMWF reference. In summary, pressure on model levels needs to be calculated using specific coefficients and surface pressure. By using pressure and temperature the altitude is derived.

P9, L210: "respectively" not correct here, either it should read "...and the lapse rate, respectively" or if you mean the vertical temperature gradient is equal to the lapse rate then it should read "The vertical temperature gradient, i.e. the lapse rate". Corrected.

P9, L211: provide the unit in the text (and based on the ACP style it should read k m-1). Thanks for pointing to this inconsistency. We now use the ACP style for the units.

P13, Figure 5: Could you add a panel showing the bias in percent? In Sect. 2.3.2 we pointed out that the selection of an adequate statistical metric is crucial for a reliable quantification of humidity errors in the UTLS. Hence, we decided to use the logarithmic formulation of the bias which is symmetrically centered around zero and not distorted after averaging (which is not the case for calculating the ratio $Q_{ERA5}$ / $Q_{DIAL}$. (see Kunz et al., 2014). Since we are aware that this log bias is more difficult to interpret, we provide the corresponding percentage value for each given value of the log bias. In addition, table 2 should help the reader.

P14, L303: Can you also add the mean bias in percent? We believe that we already had included % values in the preprint version: "*The bias increases to a maximum of +0.63 (55 %) at 1.3 km altitude above the tTP*"

P23, L439, L440 and 455: "Highest altitude" → Give here an approximate altitude. We added altitudes.

Last but not least, the original referee 2 who unfortunately could not submit a report, but thinks your study is excellent, asked why you did not use any in-situ data that was obtained during the campaigns?

We thank for providing this question. Our study focusses on the humidity profile data set from the WALES lidar, which extends previous work using comparable in situ data (Kunz et al., 2014; Kaufmann et al., 2018; Bland et al., 2021). We demonstrate that our unique data set with its large number of vertical profiles allows us to better characterize the bias in the vertical and the collocated water vapor/ozone observations indicate a possible origin. We do not think that in situ data with a limited amount of profiles would add more information for this purpose. Certainly, in situ data may be useful to pinpoint processes that are responsible for the bias in future studies.

**Technical corrections:**

P1, L14: data set → observations Corrected.

P1, L15: add "of moisture" so that it reads "vertical gradients of moisture" Corrected.

P1, L22: small → smaller (?), since you use "higher" before it should read here rather "smaller". Otherwise you could write "high" and "small". Corrected.

P2, L32: a layer → the layer Corrected.

P3, L92: from WALES? Please add. It was not clear to us how to add "from". WALES is the DIAL's name so that we think the sentence should be okay.

P4, L124: a DIAL → the DIAL Corrected.

P5, L126: Not clear. Leakage of what? In this sentence, we wanted to point to the main differences of the averaging kernels between active and passive remote sensing techniques. We tried to improve the comprehensibility of this paragraph, which now reads as follows:

"*It should be stressed, that the averaging kernel of the WALES DIAL is exactly zero outside of about $\sqrt{2}$ times the effective resolution. This is in sharp contrast to most passive remote sensing techniques where the side modes of the kernels can lead to erroneous dry or wet layers in the retrieved humidity profile.*"

P5, L140: coverage → resolution We kept coverage which we think is correct at this place. The reduced number of online wavelengths reduces the data coverage, as these wavelengths are selected to be sensitive to different water vapor concentrations and in turn to different altitude ranges. The 4-wavelength DIAL allows to measure water vapor profiles from the LS to the ground while the 2-wavelength setup (with additional ozone capability) covers only the UTLS.

P5, L144: delete "instrumented" and add "aircraft" after HALO Corrected.

P5, L157: Beyond this → However (or Additionally) Corrected.

P5, L157: are → were Corrected.

P8, L185: in 2016 → since 2016 (?) We kept "in" as the model cycles (model versions) change over time. Please see https://www.ecmwf.int/en/forecasts/documentation-and-support/changes-ecmwf-model

P8, L187: with → of the Corrected.

P8, L189: one 1 hour intervals → with a time resolution of 1 h Corrected.

P9, L229: a typical location → the typical location Corrected.

P10, L242: Rephrase sentence so that it reads "gives some example values for the bias for certain moisture observations"? Updated. Selected?

P11, Table 2 caption: Add "Some" so that it reads "Some example values" and add "according" or "respective" before "computed". Updated.

P11, L251: introduced → provided Corrected.

P11, L260: lower → smaller (?) We agree that smaller is more appropriate here.

P11, L263: add comma after model and Eq(3). Corrected.

P12, Figure 4 caption: Write "On the panels are the……….superimposed". We revised the caption in a slightly different way: *"Vertical cross sections of (a) the DIAL specific humidity (colour shading, g kg$^{-1}$), (b) ERA5 specific humidity (colour shading, g kg$^{-1}$) as well as (c) the corresponding humidity bias (colour shading) on the 1$^{st}$ October 2017. (a) – (c) are superimposed by ERA5 fields of the potential temperature (grey contours, Δθ = 3K) and the isopleths of the wind speed (magenta contours, in m s$^{-1}$), and the thermal (thick black dots) and the dynamical tropopause (2 PVU, black isoline)."*

P14, L312: add "the" or "a" before bias. Corrected.

P14, L312: compared → compared to Corrected.

P15, L326: replace "=" by "i.e." Corrected.

P19, L372: illustrated → shown Corrected.

P23, L461: delete "the" before Dyroff et al. Corrected.

---

## Author Comment (AC2)

**ACP-2022-505 – Vertical structure of the lower-stratospheric bias in the ERA5 reanalysis and its relation to mixing processes**

By Krüger et al. (2022)

Reply to review #2

This is an excellent evaluation of ERA5 water vapor using a large airborne dataset from HALO. The techniques used to assess biases are varied and comprehensive and complement existing assessments for other reanalysis models. Overall, I find the manuscript to be in great shape and have only a handful of what I hope are helpful suggestions to the authors as they work to finalize the paper.

We are grateful to reviewer #2 for the positive review and the recognition of our study. Below, we answer each comment in blue font.

**SPECIFIC COMMENTS**

1. Lines 97-102: While these outlines have become unfortunately common, I find them to be absolutely unnecessary. Recommend removing

We decided that we want to keep the outline to give an overview about the structure of the paper.

2. The maximum in humidity bias in the lower stratosphere is highlighted throughout and first introduced in Figure 5. In considering this bias and the accompanying discussion, the thought occurred to me that temperature in that layer was not evaluated in great detail. Are their sufficient temperature profile data in the HALO measurements to also assess temperature biases in these environments? That seems to be incredibly important to understanding the context for such humidity biases. Perhaps this layer, commonly characterized by containing a strong tropopause inversion layer (of similar shape to the humidity bias even), is driven in part by a warm bias in the model? For these reasons, if possible, I would strongly suggest the authors evaluate temperature bias and add that here to provide further context on the likely nature of this bias (and its variability between environments).

Unfortunately, our data set lacks collocated temperature profile data, which made it necessary to rely on simulated tropopause altitude only (see also the second comment by reviewer#1). There are a few dropsondes available for some of the considered campaigns, however, these are not representative for the entire data set. For this reason, we focussed on the LS moist bias and its relation to mixing using the comprehensive and unique DIAL data set. Certainly, a consideration of temperature observations to investigate the relation of temperature and humidity biases in the UTLS is an interesting task (see discussion in the Sect.1). Please note that recently Bland et al. (2021) used radiosonde data during the NAWDEX campaign to quantify the LS moist and its relation to temperature biases. They show that the analysis temperature representation is fairly good, but radiative effects in relation to the LS moist bias cause a cold bias that is intensifying with increasing forecast lead time.

**Technical Edits:**

Line 19: delete "located" Corrected.

Line 193: delete "on" Updated.

Line 268: a word appears to be missing here. I think the authors meant to write "the systematic **nature** of the diagnosed" Updated. Included **"nature"** in the sentence.

Line 461: delete "the" Corrected.

Line 473: "This supported" should be "This is supported" Corrected.

Line 514: "profile" should be "profiles" Corrected.

---

## Author Comment (AC3)

This paper evaluates the lower-stratospheric moist bias in a NWP reanalysis. A moist bias is a common problem in NWP and climate models in both the analysis and forecasts, as documented by others, but the novelty of this paper lies in the use of a large dataset of lidar data from aircraft observational campaigns covering several seasons and different years, and characterising well the vertical profile of the error in the latest reanalysis from ECMWF (ERA5). This is very relevant as the lower-stratospheric moist bias leads to a significant temperature error through radiative cooling, which has potential impact on the atmospheric circulation. Importantly, the paper largely confirms the results of other studies on the ECMWF (re)analysis that use other data sources, showing a significant bias in the lowermost stratosphere (LS) but additionally it quantifyies the reducing bias in the upper part of the LS, and uses ozone and water vapour to show the largest error is in the mixed tropospheric-stratospheric air layer suggesting the source of the problem is too much mixing of water vapour across the tropopause.

It is a very well written paper and excellent analysis of the DIAL data and evaluation of the ERA5 moist bias. I just have a few points to investigate and a few minor suggestions for text edits that need to be considered before publication.

We would like to thank the reviewer for the positive evaluation of our manuscript and for the valuable and constructive comments that helped us to improve the manuscript. Below, we answer each comment using a blue font.

**SPECIFIC COMMENTS**

**1. Observational errors**

Although the precision of the high quality observational data is discussed in Section 2.1 and various studies referenced, a quantitative description of the measurement uncertainty in this paper is missing, either in percentage terms or absolute specific humidities, as well as a discussion of any possible sources of bias in the observations. Although this information may be adequately described in other papers, it is important to review the estimated observational error here to be confident about the evaluation.

We follow the reviewer's suggestion and aimed to improve the discussion of data quality and possible implications for our results. As discussed in the manuscript one has to distinguish statistical (precision) and systematic errors (accuracy) that may affect the data quality. The statistical error is not expected to cause a bias in the observed humidity. We improved the discussion of error sources and quantified the estimated (total) error in the UTLS. The paragraph within Sec. 2.1 was changed to:

*"In the DIAL data retrieval, the statistical error of the observed volume is different for each flight and depends on the water vapour distribution and the background light. To remove high noise, typically occurring in dry air lying underneath moist air, e.g., in the vicinity of stratospheric intrusions (Trickl et al., 2016), we filtered 5 % of the noisiest data for each individual flight. This threshold turned out to be useful, however, reduced the data availability in the lower-to-mid troposphere. Furthermore, Rayleigh-Doppler beam broadening, laser spectral impurity and uncertainties in spectral databases are sources for systematic errors, which are compensated for in the retrieval algorithm. The total systematic error was found to be in the order of 5 % (Kiemle et al., 2008). The high reliability of WALES was*

*demonstrated in various intercomparisons, e.g., with Lyman-alpha in situ hygrometers (Kiemle et al., 2008), comparable airborne and ground-based DIAL instruments (Bhawar et al., 2011) and radiosondes with a frost point hygrometer (Trickl et al., 2016)."*

In summary, the expected error of the DIAL water vapor profiles is about an order of magnitude smaller than the diagnosed lower-stratospheric moist bias. We agree with the reviewer that possible implications of observational errors were not discussed and therefore we added to Sect. 4:

*"Furthermore, the magnitude of the LS moist bias exceeds the expected error of the DIAL humidity observations by approx. one order of magnitude which underlines the significance of our results."*

**2. Moist bias in the troposphere - Figure 5 and elsewhere**

Could the small positive bias in the troposphere be due (or partly due) to a systematic shift between the observed and ERA5 tropopause? Bland et al. (2022) shows the height of the thermal tropopause in the operational ECMWF analyses is on average 200m higher than in the radiosonde observations and this might also apply to the ERA5 reanalyses? In tropopause relative coordinates (using the ERA5 derived tropopause), this would then lead to an apparent moist bias given the significant vertical gradient in the troposphere. Is it negligible? Can you quantify this? Is there possibility of a bias in the DIAL data? These need to be discounted to be sure that the error is in the re-analysis.

We thank the reviewer for this valuable comment that we picked up in the discussion of our results. In contrast to Bland et al. we do not have collocated temperature profiles available and, thus cannot evaluate a systematic tropopause altitude bias for our data set.

To evaluate the impact of such a 200-m shift we reproduced Fig. 5 by lowering the ERA5 tropopause altitude by 200 m (see below). Indeed, the tropospheric bias is significantly reduced with the median being close to zero. The vertical structure of the lower-stratospheric bias remains comparable and although the magnitude is reduced, our results remain valid.

[Figure]

We mentioned the differences between observed and simulate tTP already in the preprint version (LL212-213) However, as we were not precise enough, we deleted this sentence and instead, we added a paragraph to the discussion in Sect. 4: *"Please note that Bland et al. (2021) show that tTP altitudes are on average about 200 m higher when derived from ECMWF IFS profiles compared to radiosondes which may impact tropopause-relative moisture distributions and in turn the bias. As no temperature observations are available, this study relies only on ERA5 tTP altitudes. Assuming a systematic shift by 200 m would reduce the tropospheric bias, however, the LS moist bias, although slightly weakened would persist."*

**3. Section 3.2.2 "Synoptic and seasonal variability".**

Figure 9 shows the synoptic variability of the vertical profile (by binning the whole dataset by tropopause altitude) and I was expecting to see a similar figure for the vertical profile binned by season. Instead Figure 8 shows the vertical profile separated by observational campaign, and then seasonal biases then has to be inferred from the specific months of each campaign. As two of the campaigns have only a small amount of data, we are also asked to disregard these profiles. Would it not be better to include a figure (or replace Fig 8) for the whole dataset binned by season to more robustly and clearly make the point about the seasonal variation of the bias? [I note that there is a sentence in lines 509-510 that refers to the similarity of two campaigns in Fig 8]

We agree and removed the NARVAL2 and EUREC4A profiles from Fig. 8. Although a plot for different seasons would be interesting, we do not think it would add much information due to the limited amount of flights. As WISE and NAWDEX represent autumn data sets and Cirrus-HL and NARVAL represent summer and winter, respectively, we think that the updated version of Fig. 8 is suitable to discuss the seasonality. Please note, that we refrain from a detailed discussion of seasonal differences of the bias due to the limited data set, which can only give some evidence for such an effect (see discussion in Sect. 4).

**TECHNICAL CORRECTIONS**

Abstract, lines 24 and 26-27 say essentially the same thing. We removed the sentence in line 24.

Line 115-116: Wording with the use of "respectively" not quite clear to say the different trace gas concentrations represent different altitude levels. Changed to: *"The online channels are sensitive to different trace gas concentrations and in turn to different altitude levels."*

Line 193: "method used on a horizontally" -> "method used a horizontally" Changed.

Line 210: Again, unusual use of the word "respectively". In this case you could use "(i.e. the lapse rate)" Done.

Line 237: You could just state this is a logarithmic formulation with base 2, just to be clear. Done.

Line 238: Something odd here with the typesetting for the equation number. Changed.

Table 2: As there is negative humidity bias in various later figures, you could add a few negative numbers in the table as well? We included one example that shows an underestimation of humidity and in turn a negative bias.

Figure 4 caption: Presumably all the derived tropopause, pot temp and wind speed is also from ERA5? State in the caption. We revised the caption of Figure 4.

Line 312: "compared the observations" -> compared to the observations" Corrected.

Line 380: "Tracer-trace" -> "tracer-tracer" Corrected.

Lines 383, 385 and Figure 11 caption: Figure 11 looks like the VMR H2O limit is 6.5ppm as the caption states, but this is inconsistent with the text on p19. Thank you for pointing to this inconsistency. Corrected to 6.5 ppm.

Could you also say why this particular value is chosen? We revised this paragraph and tried to be more precise. The selection of this values is always a bit arbitrary as e.g. discussed in Schäfler et al. (2021) and different approaches have been used. However as slight changes of the thresholds have only minor impact on the distributions in geometrical space and the percentage values in Fig. 13, we

did not go into further detail. The interested reader may find more details in the given references. We revised the entire paragraph (preprint version: pp. 19-20, LL379-389) which now reads as:

*"Following the approach by Schäfler et al. (2021), the collocated water vapour and ozone observations for four WISE flights are illustrated in tracer–tracer (T–T) phase space in Fig. 11 and three classes of observations are identified based on the characteristic distributions (e.g., Pan et al., 2004). First, tropospheric observations are characterized by low $VMR_{O3}$ (typically < 100 ppb) and a large spread of $VMR_{H2O}$. Second, high $VMR_{O3}$ at low $VMR_{H2O}$ (< 6.5 ppm or < $4x10^{-3}$ g $kg^{-1}$) are assigned to lower stratospheric air. Additionally, a class with intermediate chemical characteristics ($VMR_{H2O}$ > 6.5 ppm and $VMR_{O3}$ > 100 ppb) is attributed to mixed air masses that experienced mixing between the troposphere and stratosphere."*

Lines 416-417: Although the average vertical profile from the WISE flights (Fig 13b) is similar to the full dataset (Fig 5b) in the stratosphere, the tropospheric profile looks a bit different, i.e. fairly constant in the full dataset (0.2-0.25), but decreasing with increasing altitude in the WISE data (0.4-0.1). Please reword.

We agree that this sentence on the UT bias could be more precise and adjusted the wording based on the suggestion. The sentence reads as follows:

*"The average vertical profile of the moist bias for the WISE flights (Fig. 13a) is similar to the full dataset (Fig. 5b) at the tTP and in the LS, i.e., a local minimum is found at the tTP (0.1; 7 %) and a pronounced maximum of 0.62 (54 %) peaking at about 1 km above the tTP. The tropospheric part of the profile, however, is almost constant in the full dataset (0.2–0.25) but decreasing with increasing altitude in the WISE data (0.4–0.1)."*

Line 473: "This supported" -> "This is supported" Corrected.

Line 493 and Line 94: Research question 1 has slightly different wording here compared to the Introduction, but should be the same. Perhaps a more succinct sentence to consider is: "1. Can the mutli-campaign DIAL data set robustly quantify the LS moisture bias in ERA5". We agree and took the revised wording from the introduction.

Line 498: "The flights that were performed in different times of the year can reproduce seasonal differences in the observed humidity distributions". They rather "show" or "highlight" the seasonal differences than "reproduce" them. Changed to *"indicate"*

Line 501: "What is of the vertical" -> "What is the vertical". Corrected

Line 502: "moist bias is also contained in" Suggest, "moist bias is present in" Corrected.

Line 523: "In future" -> "In the future" Corrected